



# Modelling below-cloud scavenging of size resolved particles in GEM-MACHv3.1

Roya Ghahreman, Wanmin Gong, Paul A. Makar, Alexandru Lupu, Amanda Cole, Kulbir Banwait, Colin Lee, Ayodeji Akingunola

Environment and Climate Change Canada, Toronto, Canada

*Correspondence to*: Roya Ghahreman (roya.ghahreman@ec.gc.ca)

**Abstract.** Below-cloud scavenging is the process of aerosol removal from the atmosphere between cloud-base and the ground by precipitation (e.g. rain or snow), and affects aerosol number/mass concentrations, lifetime and distributions. An accurate representation of precipitation phases is important in treating below-cloud scavenging as

the efficiency of aerosol scavenging differs significantly between liquid and solid precipitation. The impact of different representations of below-cloud scavenging on existing model biases (Makar et al, 2018), was examined through implementing a new aerosol below-cloud scavenging scheme (from Wang et al., 2014) and comparing with the GEM-MACH's existing scavenging scheme, based on Slinn (1984). Further, the current GEM-MACH employs a single-phase precipitation for below-cloud scavenging: total precipitation is treated as either liquid or solid

depending on a fixed environment temperature threshold. Here, we consider co-existing liquid and solid precipitation phases as they are predicted by the GEM microphysics. GEM-MACH simulations are compared with - observed precipitation samples, with a focus on the particulate base cation $NH_4^+$, acidic anions $NO_3^-$, $SO_4^=$, $HSO_3^-$ in precipitation, and ambient particulate sulfate, ammonium and nitrate.

Overall, the precipitation-phase partitioning and Wang et al. (2014) scavenging scheme improve GEM-MACH

performance relative to earlier approaches. Including multi-phase approach leads to a decrease in $SO_4^{2-}$ scavenging and impacts the below-cloud scavenging of $SO_2$ into the aqueous phase over the domain. Sulphate biases improved from +46% to -5% relative to Alberta Precipitation Quality Monitoring Program wet sulphate observations. At Canadian Air and Precipitation Monitoring Network stations the biases became more negative, from -10% to -30% for the tests carried out here. These may be compared to previously published annual average biases of +200% for

$SO_4^{2-}$ from earlier versions of GEM-MACH (Makar et al, 2018), indicating an overall improvement in model performance relative to the prior results. Improvements in model performance (via scores for correlation coefficient, normalized mean bias, and/or fractional number of model values within a factor of two of observations) could also be seen, between the base case and the two simulations based on multiphase partitioning for $NO_3^-$, $NH_4^+$, and $SO_4^2$. Whether or not these improvements corresponded to increases or decreases of $NO_3^-$ and $NH_4^+$ wet deposition varied

over the simulation region. The changes were episodic in nature – the most significant changes in wet deposition were likely at specific geographic locations and represent specific cloud precipitation events. The aerosol scavenging rates of the two schemes differ during liquid precipitation in the size range of 0.1-1 µm, mostly at high precipitation intensity. The two schemes aerosol scavenging diverges for aerosols smaller than 1 µm for solid precipitation at lower intensity (R=0.01 mm/h), while at higher precipitation intensities (R=10 mm/h), the two





schemes show bigger differences for aerosols larger than 1 µm. The changes in wet scavenging resulted a higher formation rate and larger concentrations of atmospheric particle sulphate.

## 1 Introduction

Atmospheric aerosols impact air quality, and influence climate directly by absorbing and scattering sunlight, and indirectly by modifying cloud properties (e.g. Haywood and Boucher, 2000). Aerosol number/mass concentration, lifetime and distribution are regulated by the removal processes such as wet scavenging (Pruppacher and Klett, 1997). Wet scavenging, including in-cloud and below-cloud scavenging, is defined as the removal of both aerosol particles and gases from the atmosphere by different hydrometeors such as rain or snow. Scavenging of pollutants by cloud droplets and by removal due to precipitation formation (also known as "rainout") occurs as a result of the solubility of gases and via aerosol activation. Below-cloud scavenging is also known as the "washout" process, in which atmospheric material is exposed to falling hydrometeors and hence may be subject to collection and removal (e.g., Rogers and Yau, 1989). Wet scavenging is the dominant removal pathway of aerosol particles at the global scale (Stier et al., 2005), though the relative importance of aerosol dry deposition may have increased as a result of new observational studies (Emerson et al., 2020). In general, the study of the wet deposition process requires an understanding of cloud processes and precipitation chemistry, and details of these processes underlie some of the major sources of uncertainty in wet deposition modeling (Tost et al., 2007; Kajino and Aikawa, 2015). To study the below-cloud scavenging, it is necessary to consider some of the physical processes, such as condensation, Brownian motion, thermophoresis, diffusiophoresis, turbulent inertial interception, and gravitational/electric forces (Pruppacher and Klett 1978, Gong et al., 2003 and 2011). Brownian motion is more important for scavenging particles smaller than 0.1 $\mu$m, while the gravitational forces are more efficient to remove particles larger than 1 $\mu$m. Collision efficiency of hydrometeors with particles has a minimum for particles in the size-range of 0.1–2 $\mu$m. This region of low collection efficiency due to Brownian motion and gravitational settling, with its minima in this size range, is referred to as the Greenfield gap (Greenfield, 1957; Slinn and Hales, 1971; Andronache et al. 2006; Ladino, et al. 2011). The size distributions of aerosol particles and the diffusivity/solubility of gases also need to be specified in order to accurately represent the collection kernel for hydrometeor-aerosol collision efficiency (c.f. Cherrier et al, 2017). In this study, we focus on this process, the below-cloud wet scavenging of aerosols.

The below-cloud scavenging of particles is more efficient in the cold months due to snow scavenging, while gas scavenging is more efficient during the warm months (Cheng et al., 2017). Snow scavenging is an important removal mechanism in mid-latitude, polar regions and mountainous areas. Snow and ice crystals are more efficient scavengers than water drops, due to their larger surface-to-volume ratio (Wang et al., 2014). The study of snow scavenging is more complicated than rain scavenging, due to the wide variety of snow-particle shapes, sizes, and densities, which results in different fall speeds, cross-sectional areas, and flow patterns around snow particles (Pruppacher and Klett, 1997; Jylhä, 1999, Wang et al., 2014), while rain droplets are usually assumed to be spherical in shape and have a common density. The modeling study of Croft et al., (2009) estimated around 30% of below-cloud scavenging of sulphate particles is due to snow scavenging, globally.



Some studies indicate a tendency of the precipitation phase to be moving towards the liquid (rain) due to the global warming (Trenberth, 2011). It is important to differentiate precipitation phases in model simulations, as scavenging efficiency differs between rain and snow (examples of aerosol scavenging coefficients for rain and snow are shown in Figure 3). The temperature cross-over characterizing the transition point between equal contributions of rain and snow towards particle collection also varies significantly, between -0.4 and 2.4 °C (a 29-year Northern Hemisphere
observational dataset demonstrating this range may be found in Jennings et al., (2018)). Despite this temperature variation, and the importance of precipitation phase on wet scavenging and wet fluxes, some scavenging models use partition precipitation phase based on a simple, uniform air temperature threshold (Harpold, et al., 2017; Feiccabrino et al., 2015; Jennings, et al., 2018).

The rate of change (loss) of aerosol mass concentrations due to below-cloud scavenging by precipitation (rain and
snow) is referred as the scavenging coefficient Λ (s⁻¹) (Seinfeld and Pandis, 2006):

$$\frac{\partial C(t)}{\partial t} = -\Lambda \cdot C(t), \tag{1}$$

$$\Lambda(d_p) = \int_0^\infty \frac{\pi}{4} D_p^2 U_t(D_p) E(D_p, d_p) N(D_p) dD_p \tag{2}$$

In Eq. 1, $C(t)$ is the mass concentration of aerosol at time $t$, and the scavenging coefficient Λ is the loss frequency (s⁻¹) of mass concentration per unit time; their product in (1) is the rate of mass concentration loss per unit time associated
with scavenging. In Eq. 2, $d_p$, $D_p$ and $N(D_p)$ refer to aerosol diameter, droplet diameter and droplet number density at given droplet size, respectively. In this equation, $\Lambda(d_p)$ is expressed as a function of the particle collection efficiency, $E(d_p, D_d)$ and the hydrometeor's fall speed $U_t(D_d)$.

Eq. (1) is also dependent on the size of the aerosol (via $d_p$), hence requiring the aerosol size distribution to be incorporated into the equations. The modal or bulk representation of the scavenging coefficient includes the aerosol
size distribution as a superposition of lognormal subdistributions or modes with different sizes and chemical compositions (Whitby and McMurry, 1997). The modal approach is used in some previous modelling studies (e.g., Seigneur et al., 1986; Binkowski and Shankar, 1995; Zhang et al., 1999; Vignati et al., 2004; Byun and Schere, 2006; Textor et al., 2006; Zhang, 2008; Kukkonen et al., 2012; Wang et al., 2014).

Here, a semi-empirical, size-resolved Λ parameterization from Wang et al., (2014) was implemented in the
Environment and Climate Change Canada (ECCC) air quality prediction model GEM-MACH v3.1 (Global Environmental Multi-scale model–Modeling Air quality and Chemistry). The Λ parameterization in Wang et al., (2014) study was developed based on the uncertainty analyses and follows power law relationships with precipitation intensity R (Wang et al., 2014). Furthermore, it is applicable to below-cloud scavenging of the rain and snow and over a wide range of particle sizes and precipitation intensities.

In the present study, we determine the impact of precipitation phase partitioning on below-cloud aerosol scavenging, and to compare the Wang et al., (2014) scavenging scheme with the previous GEM-MACH scavenging scheme (based on Slinn 1984). The implementation details of both schemes are described below.

In what follows, below-cloud scavenging in the GEM-MACH model and the simulation set-up are described (Sections 2.1 – 2.3), followed by a brief description of the measurement data used for model evaluation (Section 2.4). Section 3



presents the results including (1) GEM-MACH below-cloud scavenging tests to examine the impact of precipitation partitioning and of the Wang et al., (2014) scheme, and (2) comparison of GEM-MACH simulations with observations. The latter work has a focus on the particulate base cation $NH_4^+$, acidic anions $NO_3^-$, $SO_4^=$ and $HSO_3^-$, and particulate sulphate, ammonium and nitrate. The summary and conclusions of our work are reported in Section 4.

## 2    Methodology

### 2.1   GEM-MACH Model

The GEM-MACH base model consists of an online tropospheric chemistry module embedded within ECCC's GEM numerical weather forecast model (Côté *et al.* 1998a, b; Charron *et al.*, 2012, Moran et al., 2013). A fully-coupled version of GEM-MACH is used for this study. The fully-coupled version allows for the model to include the influence of on-line aerosols on modelled meteorology through radiation and cloud microphysics (Makar *et al.*, 2015a, b; Gong

*et al.,* 2015; Makar *et al*., 2021). The aerosols predicted by the chemistry module are used to calculate aerosol extinction in the GEM radiation code and droplet nucleation in the GEM microphysics code, with the resulting cloud liquid water content and cloud droplet number concentration (CDNC) in turn affecting aqueous phase chemical processes within the model's chemistry modules.

The GEM-MACH chemistry module includes a comprehensive representation of air quality processes, such as gas-

phase, aqueous-phase, and heterogeneous chemistry and aerosol processes (e.g. Moran *et al.*, 2013; Makar *et al.*, 2015a, b; Gong *et al.*, 2015). The default gas-phase chemistry of GEM-MACH was based on the ADOM-II mechanism with 47 species and 114 reactions (Lurmann et al., 1986; Stockwell and Lurmann, 1989). In the current study, we use gas-phase chemistry parameterized by the SAPRC 11 mechanism with 175 species and 837 reactions (Carter and Heo, 2013); inorganic heterogeneous chemistry  represented by a modified version of ISORROPIA algorithm of Nenes *et*

*al.* (1999) as described in Makar *et al.* (2003); secondary organic aerosol (SOA) formation  parameterized using a two-product, overall or instantaneous aerosol yield formation (Odum *et al.*, 1996; Jiang 2003; Stroud *et al.*, 2018); aerosol microphysical processes, including nucleation and condensation (sulphate and SOA), hygroscopic growth, coagulation, and dry deposition/sedimentation  parameterized based on Gong *et al.* (2003); and the representation of cloud processing of gases and aerosols includes uptake and activation, aqueous-phase chemistry, and wet removal

(Gong *et al.*, 2006, 2015). The default GEM-MACH model includes eight internally mixed aerosol components: sulfate, nitrate, ammonium, primary organic aerosol, secondary organic aerosol, elemental carbon, crustal material, and sea salt. For this study, crustal material is further speciated into 6 elements: calcium, magnesium, sodium, potassium, iron, and manganese. A sectional approach is used for representing aerosol size distribution with a 12-bin (between 0.01 and 40.96 μm, logarithmically spaced) configuration.


### 2.2  Below-cloud scavenging





In GEM-MACH model, the default wet below-cloud scavenging schemes of aerosol by rain and snow are based on Slinn (1984). Wet scavenging is a removal process for the atmospheric tracers, and describes the uptake of gases and aerosols into cloud hydrometeors. Partial or complete hydrometeor evaporation subsequent to that uptake may result

in the release of captured particles back to the atmosphere. In such cases, rather than a net removal (wherein the captured material reaches the ground within precipitation), the captured material may instead be transported to a different vertical level in the atmosphere from the point where the uptake occurred. The wet deposition flux of tracer $i$ at a given level in a vertical column is calculated by Eq. (3) in GEM-MACH (Gong et al., 2006):

$$F_i(z) = (F_i(z + 1) + \Delta F_i(z))(1.0 - f_{evp}(z)) \tag{3}$$

where $f_{evp}$ is the fraction (0 to 1) of precipitation loss by evaporation. $\Delta F_i(z)$ is the input flux of aerosols to the hydrometeors at level z due to either precipitation production (e.g., cloud-to-rain) or precipitation scavenging, and $F_i(z + 1)$ is the flux arriving at that level from above.

Slinn (1984) formulations are parameterizations of the scavenging coefficient theoretically described in equation (2). Based on laboratory experiments and dimensional analyses, Slinn (1984) presented separate below-cloud scavenging schemes for rain and snow as functions of precipitation rate ($P$) and mean collection efficiency ($E$) (Gong *et al.*, 2003; 2011). For rain:

$$\Lambda(d_p) = \frac{c\, P\, E(d_p, R_m)}{R_m} \tag{4}$$

where the empirical constant $c$ is 0.5, and $R_m$ is the mean drop size. The precipitation rate $P$ is also used to obtain the mean drop radius $R_m$. For snow:

$$\Lambda(d_p) = \frac{\gamma P\, E(d_p, \lambda)}{D_m} \tag{5}$$

where $\lambda$ is the characteristic capture length, $D_m$ is the characteristic length, and $\gamma$, the empirically-derived collision efficiency, is 0.6. The values of $\lambda$ and $D_m$ vary depending on temperature and snow type. The temperature dependence

of Slinn's collection efficiency E for snow (solid precipitation) includes the consideration of hydrometeor forms (or shapes), which determine the characteristic capture length, settling velocity, etc. The different forms of precipitation are found in certain temperature ranges, e.g., needle snow (-8 ℃ < T < 0 ℃), stellar snow (-25 ℃ < T < -8 ℃), and graupel (T < -25 ℃). The characteristic capture length, settling velocity, etc. are assigned with different values based on the types of solid hydrometeors (hence the temperature ranges). For both rain and snow, collection efficiency $E$, is

a linear combination of three processes: Brownian diffusion, interception, and impaction (Slinn 1984).

In the pre-existing, base-case, GEM-MACH setup, the liquid and solid precipitation fluxes predicted from the GEM meteorological module are combined, and a single phase (*either* liquid or solid) is determined by a uniform environment temperature threshold, T = 0 ˚C (e.g. rain occurs for temperatures > 0 ˚C and snow for temperatures < 0 ˚C). In this study, a multi-phase partitioning is tested, by taking into account the liquid and solid hydrometeor phases

predicted by the microphysics in the meteorological module of GEM-MACH. Using this multi-phase partitioning





approach, consistent with the GEM microphysics employed here, the *co-existence* of both liquid and solid phases can be treated (more information on this new scheme appears in section 2.3).

In addition to the above in-cloud scavenging parameterization comparisons, we compared the existing below-cloud scavenging methodology (Slinn, 1984) to a more recent semi-empirical methodology (Wang et al., 2014). In contrast

to equations (4,5) Wang *et al*. (2014) proposed separate formulae of ʌ for rain and snow, as a function of aerosol diameter and rain rate:

$$\Lambda_i = A_i R^{B_i} \tag{6}$$

where $A_i$ and $B_i$ are empirical parameters and are polynomial functions of aerosol diameter, and the subscript *i* denotes a given particle size bin. $R$ is the precipitation rate (Wang *et al*. (2014)).


### 2.3 Model setup

This study focuses on the Oil Sands (OS) region, Alberta, Canada. Figure 1 shows the model domains and Canadian Air and Precipitation Monitoring Network (CAPMoN) and Alberta Precipitation Quality Monitoring Program (APQMP) observation stations used for model wet deposition evaluation. GEM-MACH simulations were carried out

on a limited area model (LAM) domain with 2.5 km × 2.5 km (red) resolution, nested from a 10 km × 10 km (blue) horizontal resolution, for the months of April and July, 2018. The simulations incorporated a one-week "spin-up" period prior to April and July 1$^{st}$, 2018, to allow the model cloud fields to reach a steady-state prior to the month-long simulations used for model comparison and evaluation. The model driving meteorology was updated every 24 hours, with each high resolution run duration of 30 hours incorporating a 6 hour spin-up, for the same reason. The model set-

up with regards to meteorological piloting, chemical lateral boundary condition, anthropogenic, biogenic, and wildfire emissions are similar to those presented in (Makar *et al*., 2018, 2021), with the underlying version of GEM being updated to v5.0. The meteorology is piloted by the global GEM model and initialized daily (at 06 UTC) using the Canadian Meteorological Centre's regional objective analysis. Eighty vertical, unevenly spaced, hybrid coordinate levels were used to cover between the surface and 0.1 hPa, with the lowest terrain-following model layer located about

20 m above the surface. The feedbacks between chemistry and meteorology were enabled so that the aerosols predicted by the chemistry module were allowed to influence the model radiative transfer (through aerosol extinction) and cloud microphysics (through droplet nucleation), as described above (further details can be found in Makar et al., 2015a,b, Gong *et al.*, 2015, and Makar *et al.*, 2021).

For this study, three model runs were conducted:

1. Base run – the existing GEM-MACH setup ("base-case"): The below-cloud scavenging of aerosol is parameterized based on Slinn (1984) ("Slinn1984") as described in 2.2; a single phase precipitation was employed at a given model grid-cell: the liquid and solid precipitation fluxes at a given model grid-cell are combined to a total precipitation flux, and a uniform environment temperature threshold (T = 0 ˚C) is used to determine the precipitation phase, liquid or solid, for below-cloud scavenging (e.g. > 0 ˚C rain and < 0 ˚C

snow).





2.  Multi-phase partitioning approach (or "multi-phase"): for this experiment, the below-cloud scavenging of aerosol is also based on Slinn (1984), however, coexisting multi-phase precipitation at a given model grid is considered. In contrast to the base case, the liquid and solid precipitation fluxes are not combined prior to the below-cloud scavenging calculation. Rather, the predicted liquid and solid precipitation fluxes by the GEM microphysics are used in separate below-cloud scavenging calculations (according to the predicted phase), allowing for the co-existence of liquid and solid precipitation at a given model grid-cell for the treatment of below-cloud scavenging. In both the base-case and multi-phase approaches, the formulae of Slinn (1984; Eqs. 4 and 5) are used to determine the scavenging coefficient (for liquid and solid precipitation, respectively). The main difference between the base-case and multi-phase approaches lies in whether a precipitation is treated as a single phase (base-case), or multiple phases of precipitation (multi-phase) are considered model grid cells containing clouds. The approaches consequently differ in how the precipitation phase is determined for below-cloud scavenging, i.e. through the use of a uniform temperature threshold which determines whether the precipitation is in the form of rain or snow (base-case) or through the use of meteorological model (or parameterized) rain and snow (multi-phase). Note also that both methodologies make use of Eq. (3) to describe the flux between model layers. Figure 2 shows the average temperature and the rain and snow fluxes predicted by the GEM-MACH meteorological module for April 2018 at the model hybrid level of 0.98. The study area was relatively cold, and the average temperature is in the range (-15 to 15 °C). The model predicted both solid and liquid precipitation during the month of April, with solid precipitation dominating. There are areas (model grid-cells) where coexisting liquid and solid precipitation were predicted. With the base-case approach, all precipitation (liquid + solid) in the area where temperature was below 0°C was treated as solid phase for below-cloud scavenging. Similarly, all precipitation in the area where temperature was above 0 °C was treated as liquid phase. This is illustrated in the lower panels of Figure 2 (Fig. 2d and 2e) by masking the areas with no liquid and solid precipitation. In contrast, the new multi-phase approach treats the liquid and solid precipitation as they are predicted in the GEM microphysics when applying the Slinn parameterization for liquid and solid precipitation scavenging coefficients. The comparison between the average temperature (Fig. 2a) and the solid and liquid fluxes (Fig. 2b and 2c) from these two approaches illustrates the difference between the assigned solid/liquid precipitation based on a uniform temperature (Fig. 2d and 2e - liquid precipitation for T > 0 °C and solid for T < 0 °C) and the revised approach using the predicted solid and liquid phases from the GEM microphysics module (Fig. 2b and 2c). These results indicate the possible inconsistency between the model hydrometeor distributions generated with the use of a temperature threshold versus the "actual" precipitation phases, which may further impact the aerosol scavenging/concentration and wet fluxes. Note that Figure S1, the same plot as Fig. 2, but for July 2018, shows the existence of solid phase precipitation at high altitudes.

3.  Wang et al., (2014) scavenging scheme ("Wang2014"): this simulation also uses coexisting multi-phase precipitation similar to case 2 above, with the difference being the replacement of the Slinn (1984) formulation for the below-cloud scavenging coefficient for aerosols with the semi-empirical formulation from Wang et al., (2014). Figure 3 compares the Slinn1984 and Wang2014 scavenging coefficients as a





function of aerosol size. For liquid precipitation (Fig. 3a), the difference between the Slinn1984 and Wang2014 scavenging coefficients increases with increasing precipitation intensity. The two schemes differ the most for aerosol sizes between 0.1 to 1 µm, particularly at high precipitation intensity (Fig. 3b). For the solid precipitation at lower intensity (R=0.01 mm/h), the two schemes diverge for aerosols smaller than 1 µm, while at higher precipitation intensities (R=10 mm/h), the two schemes show higher disparity for aerosols larger than 1 µm. Collection efficiency in the Slinn formula includes the effects of the processes of Brownian diffusion, interception, and impaction. However, Slinn's formulae does not include representation of some processes such as thermophoresis or diffusiophoresis, both of which may increase the collection efficiency for particles in the size range of 0.01−1 µm (e.g., Slinn and Hales, 1971; Wang et al., 1977; Mc-Gann and Jennings, 1991; Byrne and Jennings, 1993; Pranesha and Kamra, 1997; Tripathi and Harrison, 2001; Tinsley et al., 2000; Andronache, 2004;2006; Wang et al., 2010). Wang2014 includes the consideration for these two additional processes. Jones et al (2022) showed that the thermophoresis mostly enhances the collection of accumulation mode particles (0.1 – 1 µm). This may explain the underestimation of scavenging coefficient by Slinn1984 in comparison to Wang2014 for particles below 1 µm diameter. Figure 4 shows snow scavenging coefficient magnitude versus aerosol size distribution with the intensity of 0.01 mm/h and different ambient atmospheric temperatures. In the Slinn (1984) Eq. 5 (for snow or solid precipitation), collection efficiency $E$, characteristic capture length, $\lambda$, and characteristic length, $D_m$, vary with temperature. This temperature dependence is related to the assumption of the temperature regimes associated with types of hydrometeors and their shapes and sizes. However, the Wang2014 parameterization was developed assuming ambient temperatures of 15 ℃ for rain scavenging and −10 ℃ for snow scavenging, and an ambient pressure of 1013.5 hPa for both rain and snow scavenging. This introduces some uncertainties in the Wang et al (2014) approach, when the atmospheric conditions differ from these assumptions. The assumptions result in a smaller range of changes for both rain and snow scavenging values as a function of size, generally within 10% for all particle sizes except for particle within 0.1 µm - 2.0 µm for rain scavenging, where the differences are up to 30% (Wang et al. 2014).

### 2.3 Observation for model evaluation

The observation data used for model evaluation are from several surface monitoring networks, namely, precipitation chemistry data from the Canadian Air and Precipitation Monitoring Network (CAPMoN), Alberta Precipitation Quality Monitoring Program (APQMP) and air concentration of $PM_{2.5}$ and speciated $PM_{2.5}$ data from National Air Pollution Surveillance (NAPS).

### 2.3.1 Precipitation samples

Precipitation samples were collected at five sites by APQMP, and at three sites by CAPMoN (https://www.canada.ca/en/environment-climate-change/services/air-pollution/monitoring-networks-data/canadian-air-precipitation.html). Measurement locations are shown in Figure 1. These samples are collected using wet-only





precipitation samplers. The samplers are designed to be operated only during precipitation; the sampling container lids open when the precipitation is detected by the heated precipitation sensors. The primary goal of the precipitation sample collection is analysis of the major ions. For the CAPMoN samples, the collector container was lined with a polyethylene bag which was removed, sealed, weighed, refrigerated, and shipped to the laboratory for major ion analysis. For the APQMP samples, the samples were transferred from the clean collection container to a smaller sample bottle, capped, refrigerated if stored on site, and shipped to the laboratory for analysis. The collection frequency varied between sites, with some sites collecting daily samples, others collecting weekly samples.

Using recommended methods and completeness criteria of WMO/GAW (2004, updated 2015), quality control procedures were performed by the collecting networks and precipitation-weighted mean concentrations of $SO_4^=$, $NO_3^-$ and $NH_4^+$ were calculated from the samples.

Overall, the collecting method described above tends to underestimate the total precipitation amount, due to wind and evaporative loss, and delay in lid opening relative to the commencement of precipitation, so the flux of ions derived from the samples must be corrected using independent observations of total precipitation values. CAPMoN sites were equipped with separate standard rain and snow gauges for measuring the precipitation amounts in order to carry out this correction. These standard gauges were not available on APQMP sites, and hence deposition fluxes precipitation corrections were calculated using daily precipitation depth data from the nearest meteorological station (ECCC, 2022; AAF, 2022).

### 2.3.2    Speciated and total aerosol concentrations

PM$_{2.5}$ speciation program of NAPS network includes the measurement of PM$_{2.5}$ mass and speciated PM$_{2.5}$ (e.g. sulphate, nitrate and ammonium) at the existing Canada-wide air quality monitoring sites (https://www.canada.ca/en/environment-climate-change/services/air-pollution/monitoring-networks-data/national-air-pollution-program.html). Speciated PM$_{2.5}$ samples were collected using a Partisol-Plus Model 2025-D sequential dichotomous particle samplers, by splitting the incoming PM$_{10}$ sample stream into fine and coarse fractions with a virtual impactor. The mass flow rates of the fine and coarse particle streams were maintained at 15 L min$^{-1}$ and 1.7 L min$^{-1}$, respectively. PM$_{2.5}$ speciation samples were collected using a Partisol Model 2300 sequential speciation samplers, equipped with three Harvard designed, Thermo Scientific ChemComb® cartridges. The cartridges were designed to separate PM$_{2.5}$ impactor inlets and maintain a constant flow rate of 10 L min$^{-1}$. More information regarding the data collection and analysis by NAPS is available from Dabek-Zlotorzynska *et al.* (2011).

### 3    Results and Discussion

### 3.1    Modelled wet deposition

Makar *et al.,* (2018) evaluated the model total (wet and dry) deposition against observed precipitation data from the Athabasca oil sands region, and concluded that wet deposition dominates using the modelling science available at that time. The contribution to April and July 2018 mean wet deposition from different components of sulfur and nitrogen



for the base-case experiment, are shown in Figures 5 and 6, respectively. The wet deposition fluxes include both in-cloud and below-cloud scavenging. The wet scavenging of sulphate aerosol is lower during the cold season than in the warm season. $HSO_3^-$ deposition mostly occurs close to the emission sources, while the wet deposition of the oxidized form, $SO_4^=$ is the dominant and more efficient in downwind regions. The higher monthly total precipitation

in July (Fig. 7) affects the $SO_4^=$ deposition fluxes in July (Fig. 5). Furthermore, deposition of N species is higher in July than April due to higher total precipitation (Fig. 7), agricultural activities, and increased contributions of bi-directional fluxes when the ground is unfrozen. The comparison between wet deposition of N components (Fig. 6) shows higher contributions of $NH_4^+$ scavenging, especially in agriculture-rich areas during July, where reduced N is deposited closer to the emission source. The higher $NO_3^-$ flux in winter (cold season) is likely driven by the uptake

into cloud water of higher particulate nitrate concentrations. Both the temperature dependence of particle nitrate formation (which favours colder temperatures), and the shallower boundary layer in the winter, lead to higher particulate nitrate in wintertime, which may then be scavenged below clouds. Note that the $NH_4^+$ wet flux includes contributions from scavenging of particulate $NH_4^+$ and gaseous Ammonia ($NH_3$), and $NO_3^-$ wet fluxes from both scavenging of particulate nitrate and gaseous Nitric acid ($HNO_3$).

Figure 8 shows the differences in wet deposition between the base and multi-phase experiments for $SO_3^-$, $SO_4^=$, $NO_3^-$ and $NH_4^+$, respectively. The 90% confidence interval scores for each of these difference fields are shown in Figure 9. Figure 8 shows a decrease of wet deposition of $HSO_3^-$ when multi-phase partitioning is included. Although the mean wet deposition of $SO_4^=$ is decreased for some areas, Figure 8 indicates enhancement of the scavenged sulphate particles mostly by precipitation partitioning (e.g. for multi-phase experiments). When multiphase partitioning is included

explicitly in the cloud processing parts of the model, a larger amount of solid phase precipitation occurs than would be the case using a temperature threshold. These changes in phase may result in the increase and decrease of $SO_4^=$ and $HSO_3^-$, respectively. In the model, $HSO_3^-$ is formed from $SO_2$ dissolved into liquid water (cloud droplets or rain droplet), and it is assumed that ice particles (cloud ice or snow) do not take up $SO_2$. If solid precipitation dominates, there will be less uptake (or scavenging) of $SO_2$ into precipitation. This is in contrast to the case of particle sulphate

($SO_4^=$) – the process of particle scavenging (including particle sulphate) by precipitation is more efficient for solid precipitation (snow) than liquid precipitation (rain) as illustrated in Figure 2. $NO_3^-$ and $NH_4^+$ do not experience uniform changes, and indicate both increases and decreases of wet scavenging, depending on location. The 90% confidence intervals show that the largest geographical area of the results between the two models differing at greater than 90% confidence is for $HSO_3^-$, with almost all of the difference field being significant above the 90% confidence level.

Amongst $SO_4^=$, $NO_3^-$ and $NH_4^+$, the areas of confidence level above 90% for $SO_4^=$ are relatively more extensive and consistent with the areas of increased wet deposition in Fig. 8, while the areas of confidence level above 90% for the other two ions are small suggesting that these differences are occurring more sporadically, probably linked to differences in temperature and rainfall rates, as described above.

The comparison between the Multi-phase and Wang2014 runs for April 2018 shows the impact of using different

below-cloud scavenging parameterizations, Slinn1984 vs. Wang2014 (Fig 10). The corresponding 90% confidence interval scores are shown in Figure 11. Both formulations treat $SO_2$ scavenging in the same manner; differences are associated only with particle scavenging methodology. The differences in wet deposition flux of $HSO_3^-$ between the





two runs are due in part to the response of the meteorological system to changes in CDNC location and amount, and aerosol radiative properties (i.e. to the aerosols and meteorology feedbacks). Overall, the Wang2014 scheme has slightly lower $HSO_3^-$ caused by the feedback in the model, and mixed changes of $SO_4^=$, $NO_3^-$ and $NH_4^+$. July 2018 plots (Figs S2 and S3) in general, have similar patterns to the April plots where the partitioning and Wang2014 scheme are included. Here, the regions where the differences are significant are smaller than those in Figure 11, and in contrast to Figure 9, suggesting that the differences are associated with specific precipitation events and most likely when precipitation rates are large, as shown in Figure 2.

To illustrate the distinction between the Slinn1984 and Wang2014 schemes, and to explain the differences shown in Figure 10, the scavenging coefficients based on the domain averaged precipitation intensity from the two schemes are compared in Figure S4. In Figure S4a, two different schemes (Slinn1984 and Wang2014) have relatively similar scavenging coefficients during April, but differ during July, especially for aerosols at the size range of 0.1-1 µm. The lower scavenging of the Slinn's scheme can be explained by its lack of processes such as thermophoresis, which may increase the collection efficiency for particles in the size range of $0.01-1$ µm (Jones et al., 2022). This may also explain the underestimation of scavenging coefficient from the Slinn (1984) scheme, and the differences between two schemes for particles below 1 µm (refer to section 2.3). Figure S4b indicates the difference between two scavenging schemes. This difference is reflected in Figure 10, i.e. Fig. S4b indicates a difference between the two schemes for particles smaller than 1 µm, the Wang2014 scavenging coefficient being considerably higher than Slinn1984. Given the fact that the solid precipitation is dominating in the April precipitation, we expect to see higher particle wet deposition fluxes by using Wang2014 than by using Slinn1984 scheme.

### 3.2 Comparison with observations (precipitation chemistry)

Figure 12 shows the comparisons between measured and modelled precipitation amounts and wet deposition fluxes of $SO_4^=$, $NO_3^-$ and $NH_4^+$, from the three different GEM-MACH experiments (e.g. Base-case, multiphase and Wang2014) for April. The statistical evaluations are summarized in Table 1 for the APQMP and CAPMoN sites. Sample collection occurred daily at CAPMoN sites and weekly at the APQMP sites. GEM-MACH captures the precipitation events/amounts well. The precipitation partitioning and the Wang2014 scheme do not directly affect model simulation of precipitation values. Some changes (especially local changes) are expected due to the aerosol feedbacks included in the model, however the different GEM-MACH experiments show relatively similar precipitation amounts, overall (Fig. 12a-c and Table 1). The slight distinction between the precipitation amounts of three experiments is due to the feedbacks between modeled aerosols and cloud microphysics.

Comparison of the observed $SO_4^=$ data with the simulation results (Fig. 12d-f), suggests a better agreement with observations by including the Multiphase rain-snow partitioning, and further improvement in agreement associated with the use of the Wang et al. (2014) scavenging scheme. The normalized mean bias values for the rain/snow and Wang2014 experiments are improved compared to the base case (from 0.46 to -0.05) due to precipitation partitioning. Wang2014 experiment has a better correlation (R = 0.86) and better factor 2 (Fac2 = 0.64) values compared to the other two GEM-MACH experiments (Table 1). For the CAPMoN sites, the correlation values are slightly better for





the multiphase rain-snow and Wang2014 experiments (R = 0.92 and 0.93), however, the NMB value is smaller for the base experiment (NMB = 0.10, compared to 0.27 and 0.30 for the other two runs). All of these NMB values are improved relative to the estimates by Makar et al., (2018) using a previous version of GEM-MACH, wherein wet sulphate precipitation fluxes were biased high (the slope of the linear fit of 2.2), although the low bias for the base experiment suggests this overall improvement may be due to other changes aside from wet scavenging. The latter reference suggested the high bias may reflect an underestimation of the $SO_2$ dry deposition flux closer to the oil-sands sources, and a corresponding overestimation of in sulphate particles downwind. More recent work (Hayden et al, 2021) suggests that an underestimate in modelled $SO_2$ dry deposition fluxes relative to observations may be due to previously missing impacts of co-deposition of base cations on surface pH and hence $SO_2$ dry deposition velocity. Although CAPMoN and APQMP sites are located far from the oil-sands $SO_2$ emission sources; the plots seem to indicate a positive bias only in modelled S wet flux from the base-case experiment at the APQMP sites, which has diminished in the other two experiments. Biases in S wet deposition were negative at the CAPMoN sites. We note that the current model emissions year (2018) has lower reported $SO_2$ emissions than the 2013 emissions year simulated in Makar et al (2018), though discrepancies between reported and satellite-derived $SO_2$ emissions have been noted (McLinden et al., 2020). The current model version also has higher particle dry deposition velocities than in Makar et al. (2018), following Emerson et al (2020), hence results in less particle sulphate being available for wet scavenging near the surface.

Overall, GEM-MACH estimates of wet deposited nitrogen are slightly biased low relative to observations. For both APQMP and CAPMoN, $NO_3^-$ has better correlation values (R = 0.58 and 0.76, respectively) for the multiphase and Wang2014 experiments relative to the base case. GEM-MACH experiments have relatively similar statistical scores for the APQMP $NH_4^+$ results, with the highest factor 2 value for the Wang2014 (Fac2 = 0.71). CAPMoN $NH_4^+$ shows slightly a better correlation (R = 0.69) and a better NMB for the Wang2014 experiment. Overall, the Wang2014 simulation has superior performance to the base case and multiphase Slinn1984 simulations.

### 3.3 Comparison with speciated PM data (NAPS)

The impacts of partitioning and Wang2014 scavenging on modelled ambient concentration of speciated PM2.5 (sulphate (SU), nitrate (NI) and ammonium (AM)) for April 2018 are shown in Figures 13, 14 and 15. The upper panels are April mean of SU2.5, NI2.5, and AM2.5 for the base case, and the middle left panels show the net difference between Multiphase and base-case Slinn1984 experiments, while the middle right panels show the net difference between Wang2014 and Multiphase experiments. Corresponding 90% confidence interval scores for the difference plots are shown in the lower panels. Multiphase partitioning leads to higher modelled concentration of particulate sulphate in the atmosphere, and the increases are statistically significant at the 90% confidence level. For particulate nitrate, partitioning leads to both increased and decreased concentrations, depending on location. However, based on the confidence level panels, the overall increase in particulate nitrate over the OS source area and downwind is not significant at the 90% confidence level. For particulate ammonium, the increase of the concentration associated with





the partitioning approaches is the dominant change over the entire region, while a decrease occurs near the oil-sands emissions sources.

Figure 16 and Table 2 show GEM-MACH simulation results of speciated aerosols, in the 2.5-km domain, compared with the daily NAPS observation data for April 2018. For particulate sulphate, both multiphase rain-snow and Wang2014 experiments have lower NMB compared to the base case, and the correlation value is improved for the Wang2014. Particulate nitrate model outputs show almost the same statistical evaluation results for all three experiments, and the correlation between the measured and model using Wang2014 scheme is the highest for the

ammonium. Overall the results show enhanced performance in the multiphase rain-snow experiment particularly over the source area and downwind. Model simulations making use of Wang2014 show mostly enhanced performance over OS facilities area and downwind, and a reduction over the area upwind of the OS, and these results are consistent with the changes in the wet scavenging.

### 4 Conclusions

To examine cloud processes and precipitation chemistry, we considered the co-existence of multi-phase precipitation, as predicted by the GEM microphysics, in GEM-MACH's below-cloud scavenging representation. Further, we implemented a new aerosol below-cloud scavenging scheme (Wang et al., 2014) and compared with the GEM-MACH's existing scavenging scheme, based on Slinn (1984).

An accurate representation of the precipitation phase is important in modelling the wet scavenging of atmospheric

tracers particularly in cold environment. Considering the coexistence of multi-phase precipitation in below-cloud scavenging has a more consistent impact on the precipitation scavenging of $SO_2$. Here we have noted improvements in model performance based on the model-observation performance scores associated with multi-phase partitioning and in comparison to wet deposition evaluations carried out in the region in previous work (Makar et al., 2018). For example, the multi-phase approach resulted in the most significant improvement in modelled $SO_4^=$ wet deposition flux

over Alberta (at APQMP sites, and in comparison to previously published work which had wet sulphate positive biases of +200% across combined CAPMoN and APQMP sites, Makar et al., 2018), as well as improvement in modelled ambient particulate sulfate concentration at NAPS sites.

As shown in this study and other existing studies, there is a considerable uncertainty in the various existing

parameterizations for below-cloud scavenging of aerosol particles. Of the two schemes examined, the Slinn (1984) parameterization is theoretically based and lacks representation of several physical processes involved in the particle scavenging by falling hydrometeors (such as thermophoresis and diffusiophoresis). The Wang et al., (2014) scheme is based on a semi-empirical approach, providing an overall best fit to an ensemble of existing parameterization and observations. The resulting scavenging coefficients from the two schemes show the greatest difference for aerosol

sizes between 0.1 to 1 µm, particularly at high precipitation intensity. For the solid precipitation at lower intensity, the two schemes diverge for aerosols smaller than 1 µm, while at higher precipitation intensities, the two schemes show higher disparity for aerosols larger than 1 µm. This resulted in the varied differences in modelled wet deposition (April



– winter, July – summer), especially, higher Wang2014 scavenging coefficient for particles smaller than 1 μm during the April, and higher particle wet deposition fluxes from using Wang2014 than using Slinn1984.

The model evaluation against observations (precipitation chemistry and ambient air concentration of speciated PM) seems to indicate that the use of Wang2014 parameterization along with the consideration for co-existence of multi-phase precipitation results in the best scores overall, with the most significant improvement from the multi-phase partitioning in below-cloud scavenging. The multi-phase partitioning and Wang2014 scheme improve the comparison between observation and modeled results. Comparison of the $SO_4^=$ data from the APQMP sites with the simulation

results suggests better agreement by including the multiphase rain-snow partitioning and further improvement in agreement associated with the Wang et al. (2014) scavenging scheme. GEM-MACH estimates of wet deposited nitrogen are slightly biased low, and the simulated results improve by adding the partitioning and Wang2014 scheme (relative to the base case). GEM-MACH experiments have relatively similar statistical scores for $NH_4^+$ results. Including the partitioning and Wang2014 scavenging impacts the modelled ambient concentration of speciated PM2.5

(sulphate, nitrate and ammonium). It leads to improvements in the model performance scores, with higher modelled concentration of particulate sulphate, and both increase and decrease of particulate nitrate concentration - with an overall increase over the OS source area and downwind. For particulate ammonium, the increase of the concentration is the dominant change over the entire region.

**Code Availability**

The GEM-MACH code can be downloaded from this Zenodo site: **GEM-MACH | Zenodo**. GEM-MACH model output data are available with email request from Roya Ghahreman: **roya.ghahreman@ec.gc.ca**. The model output requires a large amount of memory space in a binary format specific to Environment and Climate Change Canada's modelling systems. The conversion to other formats may be possible by on email request.


**Author contributions**

WG and RG designed the study. RG implemented the new experiments in the model under the supervision of WG and PM and with assistance from AL, CL and AA. AC and KB provided the measurement data. RG prepared the paper with contributions from all co-authors.


**Competing interests**

The authors declare that they have no conflict of interest.

**Acknowledgments**

This work was funded under the Oil Sands Monitoring (OSM) Program. It is independent of any position of the OSM Program.



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



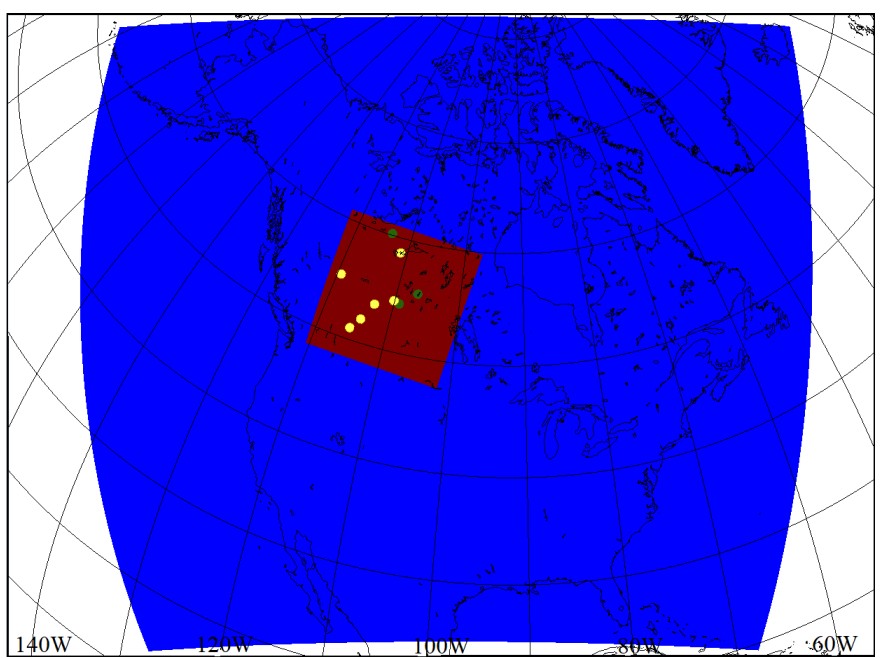

**Figure 1: Model domains - 2.5 km × 2.5 km in red and 10 km × 10 km in blue. CAPMoN (green dotes) and APQMP (yellow dotes) observation stations.**









**Figure. 2: Mean April: a. temperature, b. solid precipitation flux, b. liquid precipitation flux at the model hybrid level of 0.98. c, d: as in a, b but showing approximate limits masking out the predominantly rain flux (c) and predominantly snow flux (d). The black masked regions are the portion of the solid precipitation that would be treated as liquid in the base-case due to the above-zero environment temperature (2c), and the portion of the liquid precipitation that would be treated as solid in the base-case due to sub-zero environment temperature (2e).**











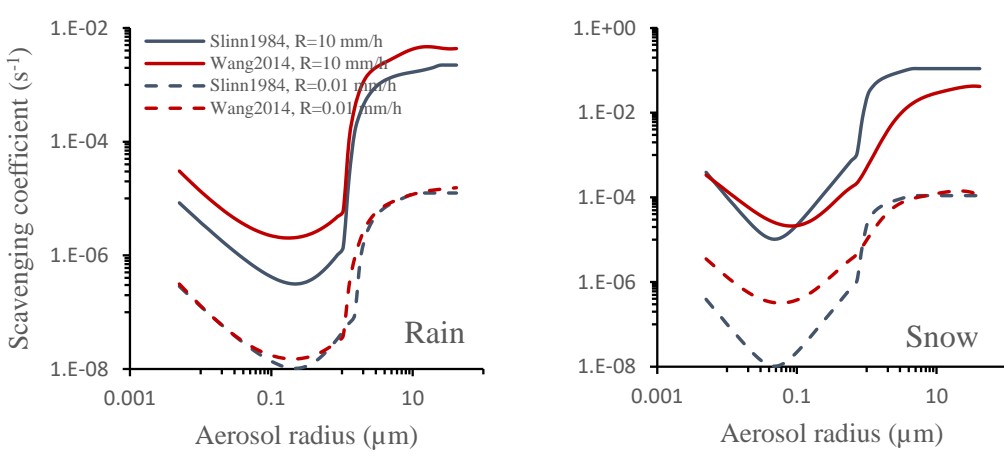

**Figure. 3: Slinn (1984) and Wang2014 rain (a) and snow (b) scavenging coefficients (s⁻¹) versus the aerosol sizes and different precipitation intensities (R).**




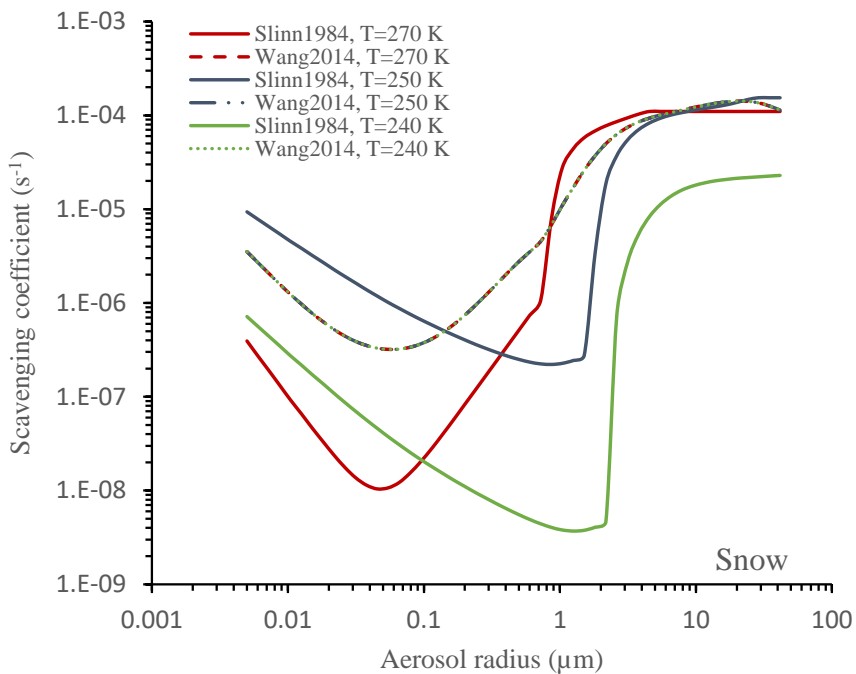

**Figure. 4: Slinn (1984) and Wang et al. (2014a) snow scavenging coefficient versus aerosol size distribution with the intensity of 0.01 mm/h at three different ambient atmospheric temperatures (T = 240, 250 and 270 K). Note that the three curves associated with Wang et al (2014a) collapse into a single curve due to their independence to temperature.**




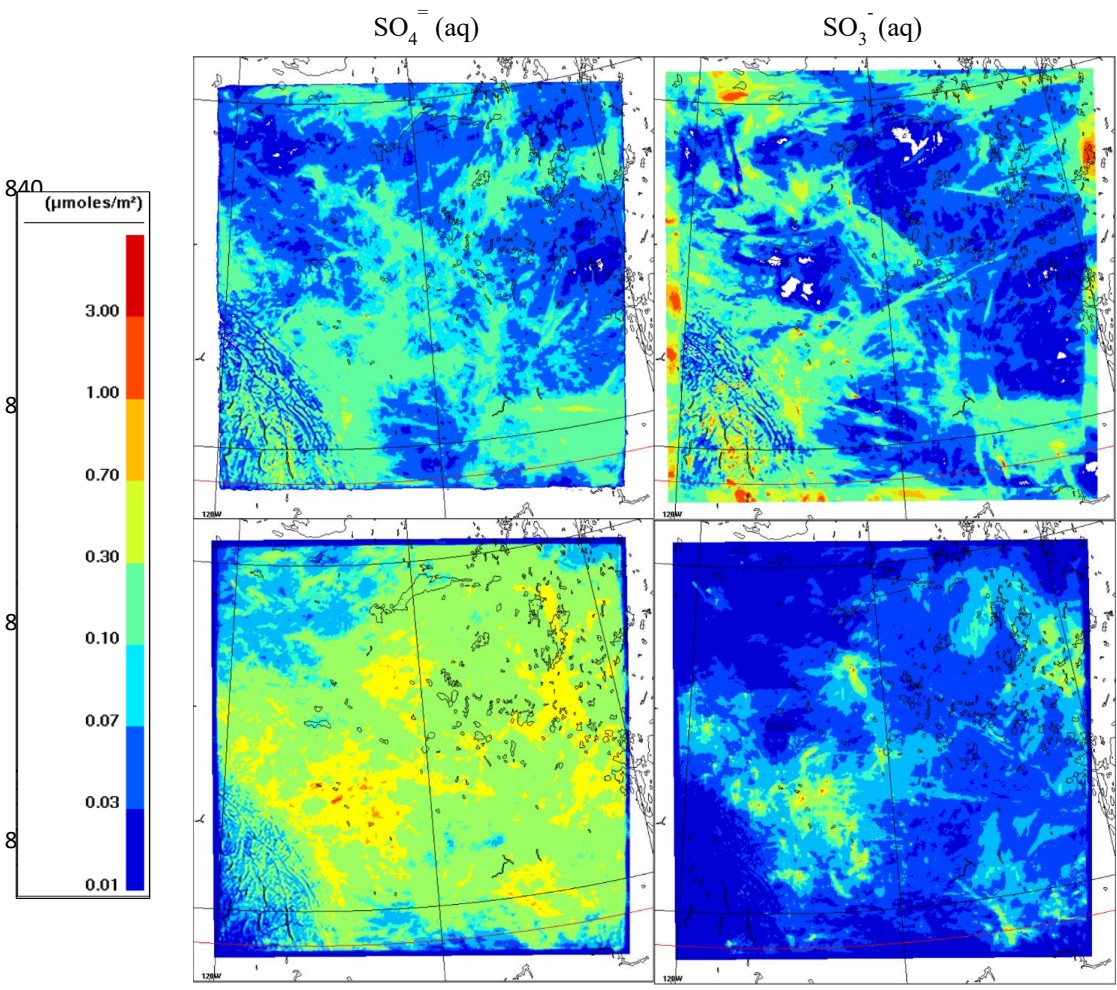

**Figure 5: Mean sulfur components concentration from the base-case experiment for April 2018 (upper panels) and July 2018 (lower panels) – SO$_4^=$ (left panels), SO$_3^-$ (right panels).**


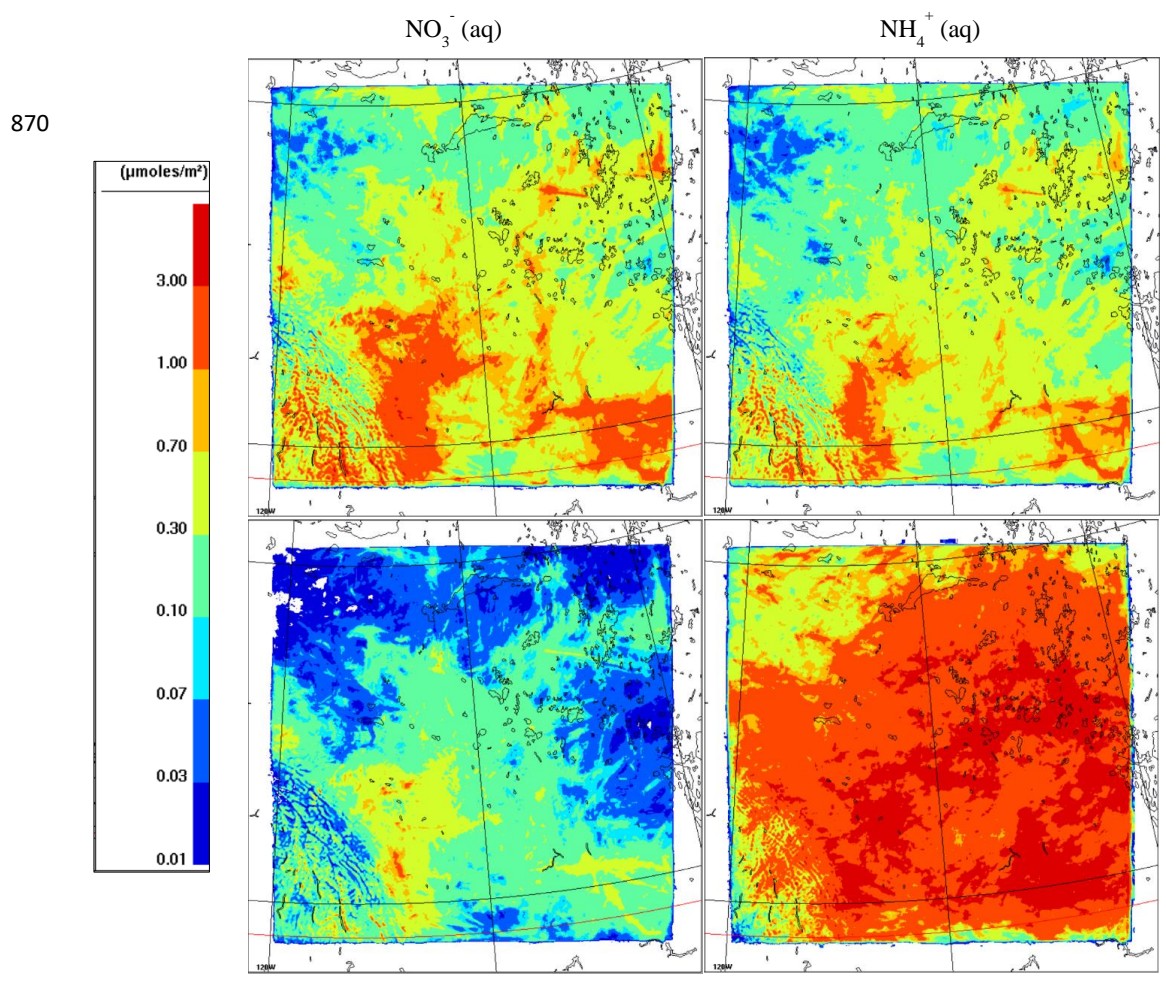


**Figure 6: Mean nitrogen components from the base-case experiment for April 2018 (upper panels) and July 2018 (lower panels) – $NO_3^-$ (left panels) and $NH_4^+$ (right panels).**







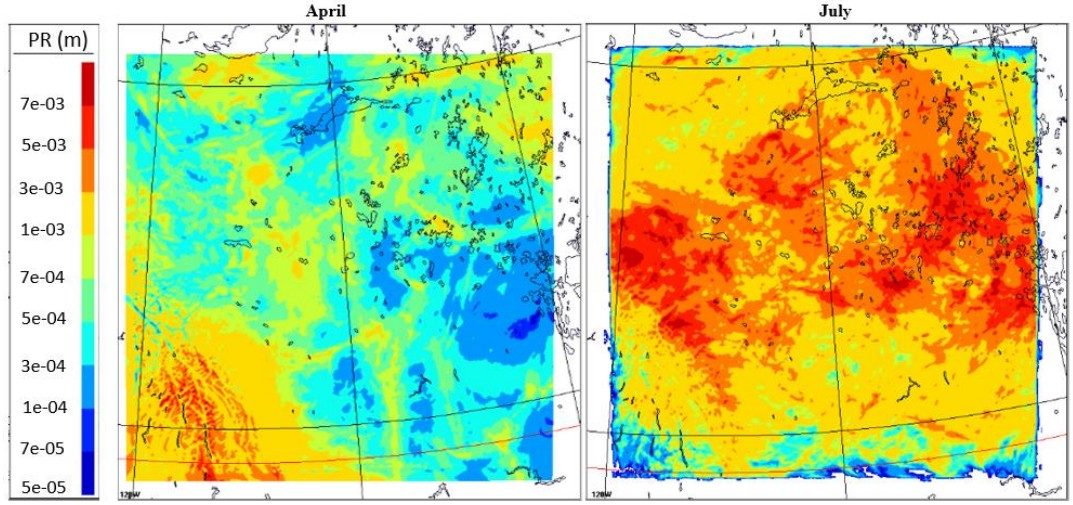

**Figure 7: Mean precipitation (PR) for April and July 2018.**





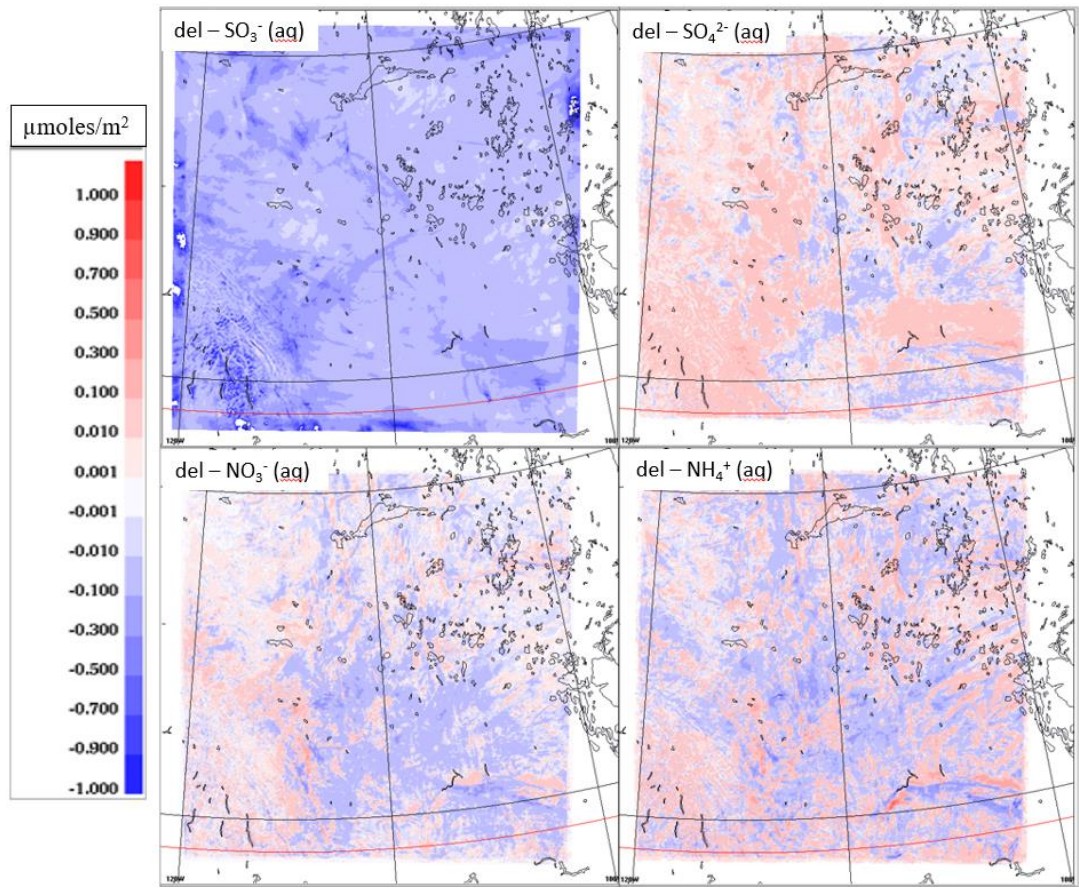

**Figure 8: The net differences of mean $SO_3^{1-}$, $SO_4^{=}$, $NH_4^+$ and $NO_3^-$ for the multi-phase and base-case experiments (e.g. multi-phase – base) for April 2018.**





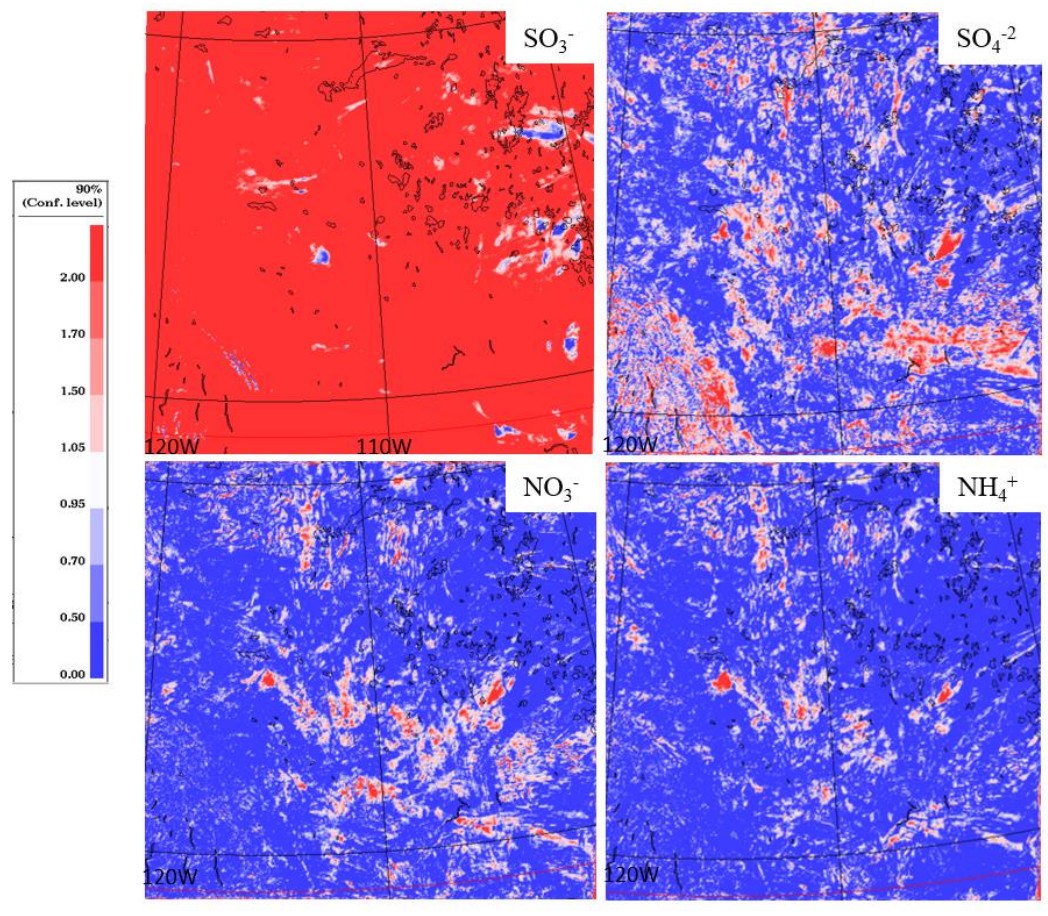

**Figure 9: The corresponding 90% confidence interval scores for Fig. 8 (following Makar et al., 2021, the differences are significant at or above 90% confidence level when the score ≥1; red regions identify > 90% confidence differences).**





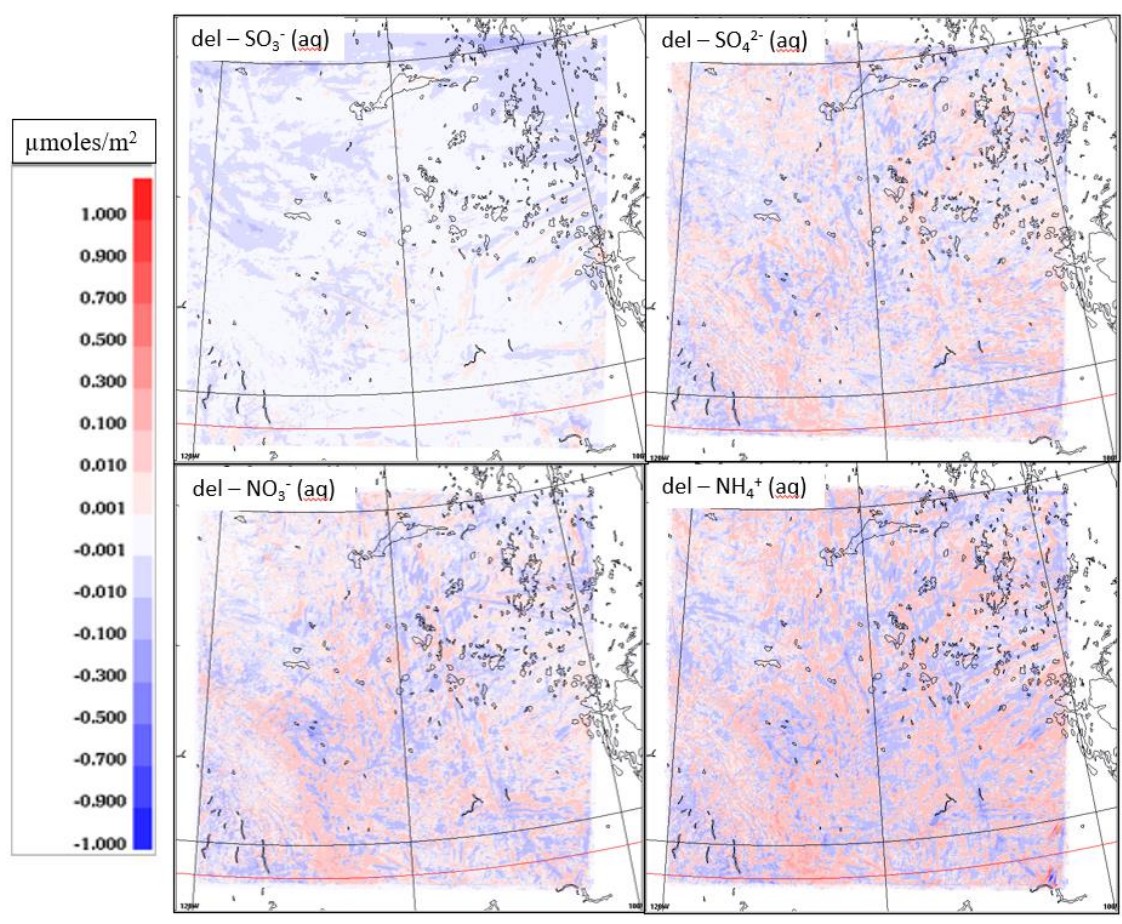

**Figure 10: The net differences of mean $SO_3^{1-}$, $SO_4^=$, $NH_4^+$ and $NO_3^-$ for the Wang2014 and multi-phase experiments (e.g. Wang2014 - multi-phase) for April 2018.**



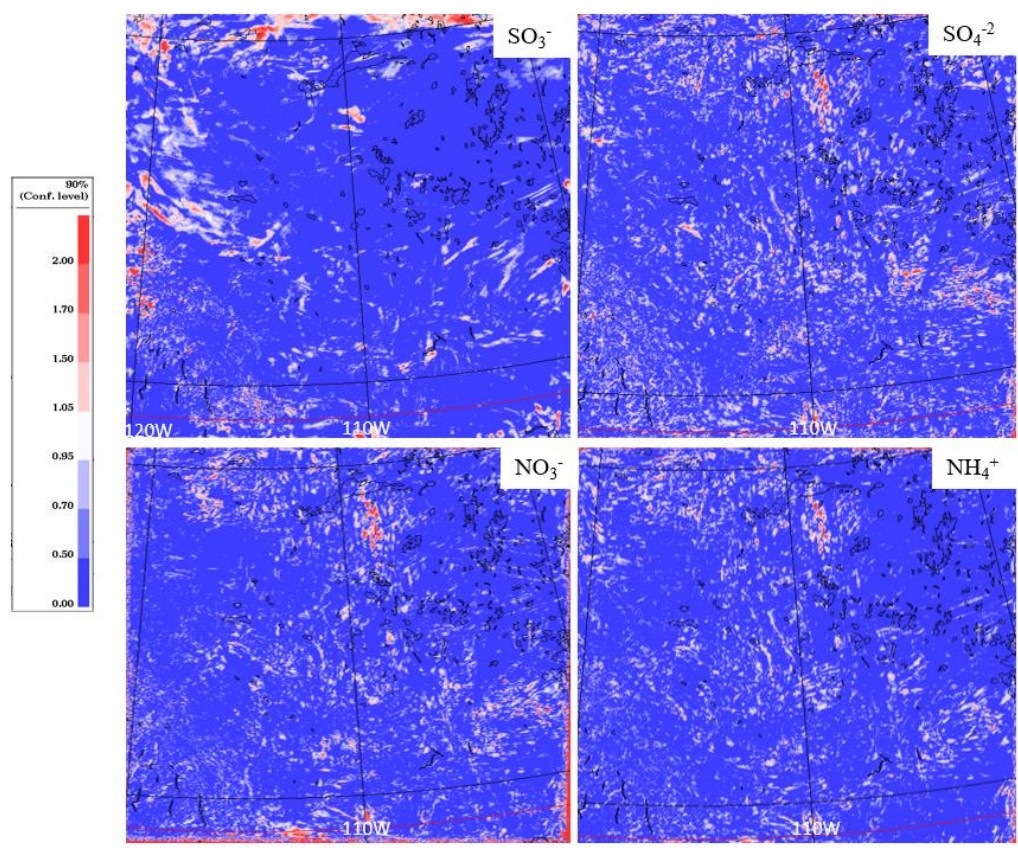

**Figure 11: The corresponding 90% confidence interval scores for Fig. 10 (following Makar et al., 2021, the differences are significant at or above 90% confidence level when the score ≥1).**



Precipitation amount (mm) – APQMP + CAPMoN

Sulphate Wet Deposition ($\mu$mol m$^{-2}$) – APQMP + CAPMoN

Nitrate Wet Deposition ($\mu$mol m$^{-2}$) – APQMP + CAPMoN


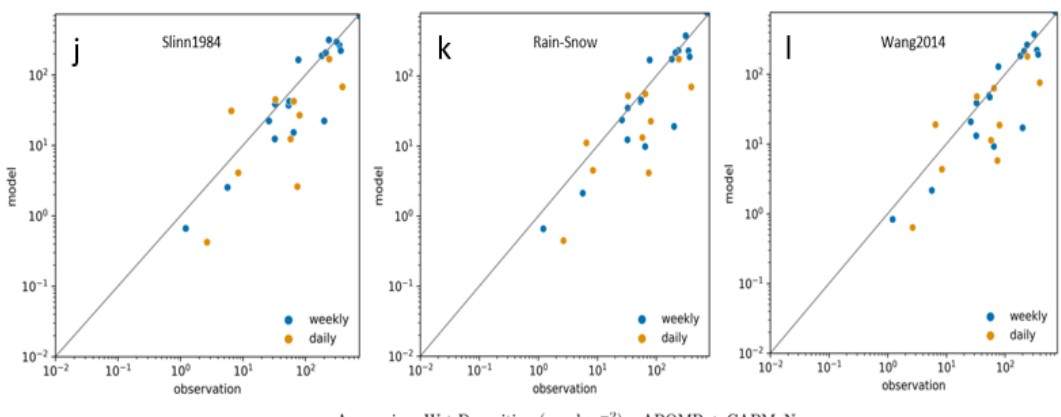

**Figure 12: Scatter plots for the precipitation amounts (a-c), sulphate (d-f), nitrate (g-i) and ammonium (j-l) wet deposition (µmol m⁻²). CAPMoN and APQMP versus GEM-MACH simulations: base run (or Slinn (1984) scheme) vs observation (left panels), rain/snow vs observation (middle panels) and Wang2014 vs observation (right panels).**










**Figure 13: April 2018 mean fine particulate sulphate (SU2.5) concentration (upper panel). The net difference between partitioned and base experiments (left, middle panel) and the net difference between Wang2014 and partitioned experiments (right, middle panel). The corresponding 90% confidence interval scores are shown in the lower panels.**








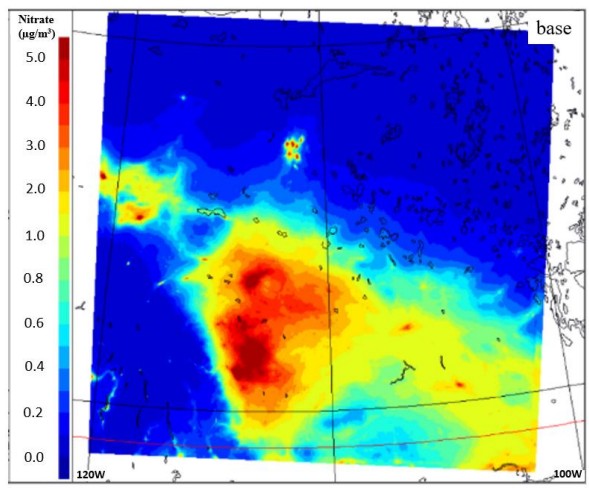


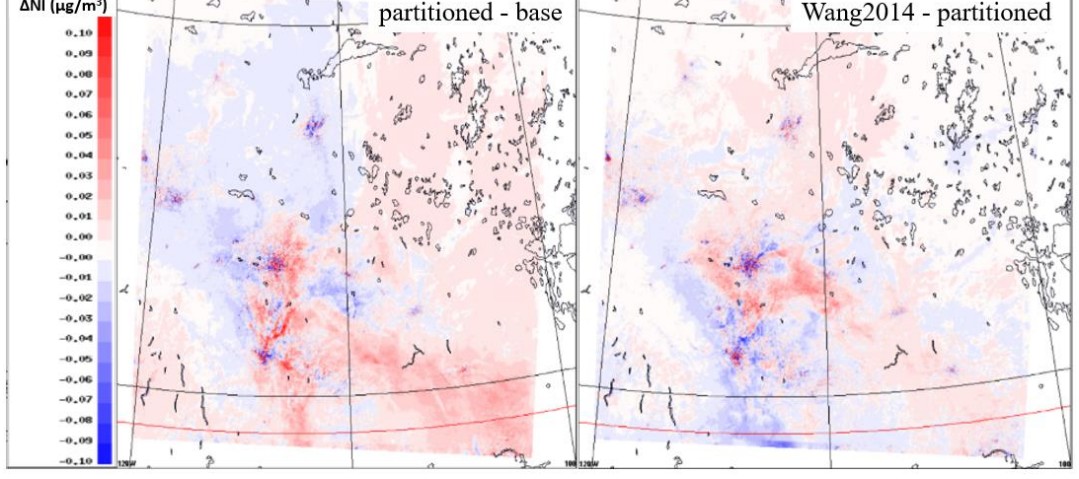

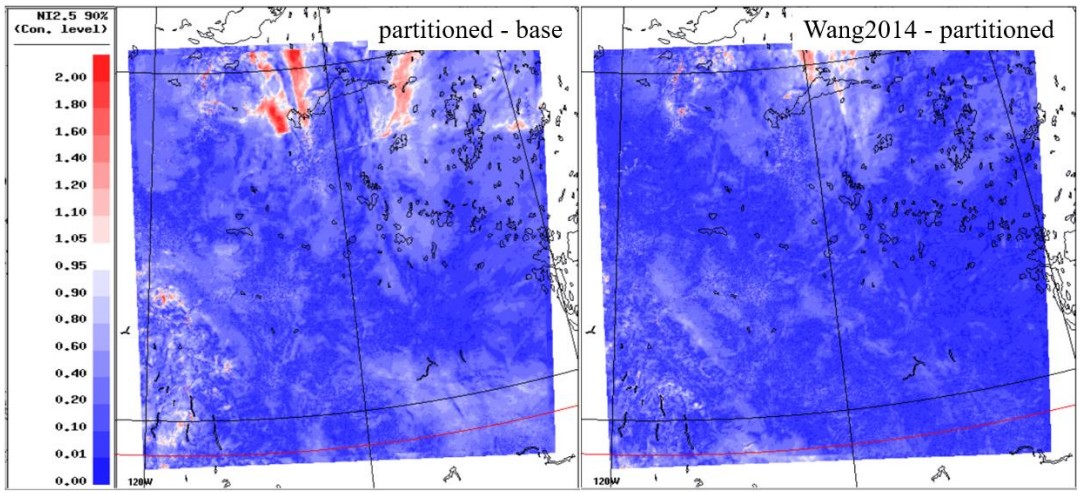





**Figure 14: April 2018 mean nitrate (PM2.5) concentration (upper panel). The net difference between partitioned and base experiments (left, lower panel) and the net difference between Wang2014 and partitioned experiments (right, lower panel). The corresponding 90% confidence interval scores are shown in the lower panels.**











**Figure 15: April 2018 mean ammonium (PM2.5) concentration (upper panel). The net difference between partitioned and base experiments (left, lower panel) and the net difference between Wang2014 and partitioned experiments (right, lower panel). The corresponding 90% confidence interval scores are shown in the lower panels.**








**Figure 16: GEM-MACH simulation results of speciated aerosols - Sulfate (upper panels, a-c), nitrate (middle panels, d-f) and Ammonium (lower panels, g-i) - compared with the daily NAPS observation data for April 2018. base run (or Slinn (1984) scheme) vs observation (left panels), multi-phase vs observation (middle panels) and Wang2014 vs observation (right panels).**




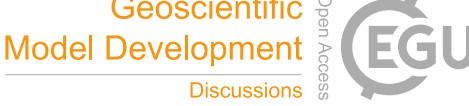

Table 1: Performance scores for the different scavenging approaches - the comparisons between the measured and modelled precipitation amounts and wet deposition fluxes of $SO_4^=$, $NO_3^-$ and $NH_4^+$. Highest scores are indicated in **bold face type**.

| | | APQMP | | | CAPMoN | | |
|---|---|---|---|---|---|---|---|
| | | R | NMB | FAC. 2 | R | NMB | FAC. 2 |
| PR | Base run | 0.73 | 0.39 | 0.29 | 0.68 | 0.20 | **0.34** |
| | multi-phase | 0.73 | **0.37** | 0.29 | **0.71** | 0.19 | 0.32 |
| | Wang2014 | 0.73 | **0.37** | 0.29 | 0.69 | **0.18** | 0.32 |
| $SO_4^{2-}$ | Base run | 0.83 | 0.46 | 0.57 | 0.90 | **-0.10** | 0.33 |
| | multi-phase | 0.84 | **-0.05** | 0.57 | 0.92 | -0.27 | **0.47** |
| | Wang2014 | **0.86** | **-0.05** | **0.64** | **0.93** | -0.30 | 0.33 |
| $NO_3^{-1}$ | Base run | 0.51 | **-0.09** | 0.43 | 0.73 | -0.68 | 0.27 |
| | multi-phase | **0.58** | -0.13 | 0.43 | **0.76** | -0.68 | **0.33** |
| | Wang2014 | **0.58** | -0.11 | **0.57** | **0.76** | -0.68 | 0.27 |
| $NH_4^{+1}$ | Base run | **0.93** | -0.14 | 0.64 | 0.68 | -0.47 | 0.40 |
| | multi-phase | 0.92 | -0.14 | 0.64 | 0.68 | -0.46 | **0.53** |
| | Wang2014 | **0.93** | -0.14 | **0.71** | **0.69** | **-0.44** | 0.47 |







Table 2: Performance scores for the different scavenging approaches - GEM-MACH simulation results of speciated aerosols, in the 2.5-km domain, compared with the daily NAPS observation data for April 2018.

|  |  | NAPS | | |
|---|---|---|---|---|
|  |  | R | NMB | FAC. 2 |
| $SO_4^{2-}$ | Base run | 0.65 | -0.41 | 0.54 |
|  | multi-phase | 0.69 | **-0.38** | **0.58** |
|  | Wang2014 | **0.70** | -0.38 | **0.58** |
| $NO_3^{-1}$ | Base run | 0.70 | 7.18 | 0.04 |
|  | multi-phase | 0.70 | 7.32 | 0.04 |
|  | Wang2014 | 0.70 | **7.07** | 0.04 |
| $NH_4^{+1}$ | Base run | 0.48 | 2.27 | 0.46 |
|  | multi-phase | 0.45 | 2.35 | 0.46 |
|  | Wang2014 | **0.50** | **2.26** | 0.46 |
