# Peer review of "Modelling below-cloud scavenging of size resolved particles in GEM-MACHv3.1"

_Geoscientific Model Development, 2023_

## Referee Comment (RC1)

**Review of Modelling below-cloud scavenging of size resolved particles in GEM-MACHv3.1 by Ghahreman et al. submitted to Atmos. Chem. Phys.**

In this paper Ghahreman et al describe changes to the treatment of below cloud scavenging in the GEM-MACHv3.1 local area model. Specifically, they test the addition of multi-phase precipitation wherein both snow and liquid rain can scavenge aerosol separately, and the replacement of the Slinn (1984) scavenging coefficients with Wang et al (2014) scavenging coefficients. They run simulations with these different models in a domain over Canada for April 2018 and July 2018 months and then compared simulated aerosol deposition rates and near-surface concentrations with 3 observation networks. They find that the addition of multi-phase scavenging and the Wang BCS model leads to large changes in SO4 deposition rates and concentrations.

At the outset, I really like the study design – a simple sensitivity test using a detailed aerosol/cloud scheme embedded in a detailed LAM and with a clean comparison against observations. I think the study design is appropriate for this investigation. Where I am less impressed with the manuscript is in the interpretation and presentation of the results. I think that the statistical methods utilised by the authors are mostly appropriate (bias, $R^2$, etc) but the statistics are somewhat misused to support a preordained conclusion that the model is improved by the new scheme. The changes to aerosol deposition rates and concentrations are on the most part marginal and therefore don't support certain statements in the abstract or conclusion. I would like to see a more balanced conclusion which reflects the insignificance of impacts on NH4 and NO3 and how some SO4 metrics are improved while others are made worse by the inclusion of the multi-phase/Wang model.

I think the paper would benefit from structural re-organization and re-writing to remove repetition and add clarification, which is reflected in the sheer number of specific comments listed below. I recommend that the paper be revised according to the many comments below before being considered for publication.

**General comments**

The abstract is too wordy and does not follow the convention of describing the problem, then the method of solving the problem, and then the key results

Why was the Wang et al (2014) model chosen to replace Slinn (1984) rather than explicitly adding the missing processes of phoreses, electric charge and rear capture which has been shown to better represent observations and empirical models than both Slinn and Wang (see e.g., Fig 3 in Jones et al., 2022)?

Figures presenting the results in LAM domain (Figs 5-11 and S1-S3) would significantly benefit from having domain-average values also presented. At present, the colour scale and the significant inner-domain variability means it is very difficult to identify the overall impact of changes – e.g., whether the net difference between Wang2014 and multi-phase in Fig. 10 b-d is negative or positive

The figure and table captions are in general not descriptive enough (in some places, outright confusing) and should at the least include the specific metric plotted (e.g., accumulated aerosol

deposition in Figs 5 and 6 which are currently erroneously labelled as concentrations). This should also be done for figures in the supplement.

Please choose informative names for the simulations and then use these names consistently throughout the manuscript.

My biggest issue with the paper is that it does not convincingly show that replacing the single-phase Slinn BCS scheme with the multi-phase Wang BCS scheme significantly improves aerosol concentrations relative to observations, which is explicitly stated as the main result of the paper in both the conclusions and the abstract. Case in point, the abstract includes the line: "Improvements in model performance (via scores for correlation coefficient, normalized mean bias, and/or fractional number of model values within a factor of two of observations) could also be seen, between the base case and the two simulations based on multiphase partitioning for NO3-, NH4+, and SO42." This is patently not true – the paper shows no improvements for NO3 or NH4 (e.g., no areas with significant changes in Fig 9c,d and insignificant improvements to only some skill scores in Tables 1 and 2). I think it is fair to say there is a significant difference in SO4 concentrations between the simulations, but in terms of improvement, SO4 is improved against APQMP observations and made worse against CAPMoN observations in terms of bias, while changes to R2 and FAC2 are marginal. I recommend that the authors be clearer about the significance of their results and include less generalised comments about how the model is improved.

**Specific comments**

[L1] The title is different to the title in the Supplement. I think the more succinct title here is the more appropriate

[L9] It is unclear what you mean by distributions – aerosol size distributions? Please be more descriptive

[L11] References to Makar et al (2018) are unnecessary in the abstract. Consider also removing references to Slinn (1984) and Wang et al (2014).

[L13] Please give the long-name for GEM-MACH here

[L16] "GEM-MACH simulations…" – please be a little bit more specific about the experiment design (e.g., Regional GEM-MACH simulations in a local-area domain over Canada)

[L17] It would help to define the chemical formulae at the outset, e.g., sulfate (SO42=) and then use the chemical formulae or the chemical name consistently throughout the paper. Bouncing between formulae and names is confusing to the reader who might think that one refers to particulates and the other is in reference to gaseous component

[L31] Sentences beginning "The aerosol scavenging rates…" and ending "…bigger differences for aerosols larger that 1um" – these are very technical statements for an abstract and should probably be moved to the model description or results. They do not add anything to the abstract. Instead, earlier in the abstract you could describe the main difference between Wang and Slinn in one sentence or at least the motivation for moving from Slinn to Wang.

[L47] – "though the relative importance of aerosol dry deposition may have increased as a result of new observational studies" – the statement is rather awkward and could be interpreted that aerosol dry deposition was directly influenced by the observational studies. Please reword

[L49] – by 'precipitation chemistry' do you mean the chemical reactions within the hydrometeor. This phrase is rather ambiguous

[L50] – when you list the microphysical processes involved in BCS you could include the rear-capture effect. I think it is well understood that rear-capture in the wake of an oblate droplet is an important BCS process (e.g., https://acp.copernicus.org/articles/17/4159/2017/) and the reason it is not often considered is that the original BCS studies considered droplets to be spherical

[L60] You say "below-cloud wet scavenging" or "below-cloud scavenging" a lot, consider acronymising this to make the manuscript easier to read

[L67] "while rain droplets are usually assumed to be spherical in shape" – it is true that most of the early BCS studies assumed spherical raindrops. It would be worth noting that this assumption is erroneous, particularly for larger raindrops, which are much more oblate

[L79 – L87] – this really doesn't belong in the abstract, please move it to the methodology section (2.2). This describes the underlying BCS scheme and does not fit well in the abstract

[L87] I find it curious that the extinction efficiency $E(dp, Dd)$ which dominates the variation in the scavenging coefficient is not described in more detail here. Potentially you could say "the Slinn (1984) extinction efficiency is prescribed as a linear combination of the extinction efficiencies due to Brownian motion, interception, and impaction. Therefore, the Slinn model misses important processes in the Greenfield gap such as thermophoresis which are included implicitly in the Wang et al (2014) model. This omission forms the motivation for testing the Wang scheme in this study".

[L87] – Are the extinction efficiencies calculated offline or online? If offline, what parameters and assumptions were made to generate the extinction efficiency (temperature, pressure, etc)? What model was used for the raindrop terminal velocity? These are important assumptions which will affect the results

[L94] – What is the motivation for testing Wang et al (2014) over Slinn? At present, it seems arbitrary

[L95] – referring to GEM-MACH, typically the acronym rather than the long name is in parentheses. Also, this should be defined at the first use of the model's acronym

[L112] "fully-coupled" with reference to GLMs or LAMs typically refers to atmosphere-ocean coupling unless otherwise specified, please be more specific about what is fully coupled to what

[L130] Sentence beginning "The default GEM-MACH model includes…" - This seems like a very comprehensive aerosol and chemistry model. Has it been used for other simulations - is it validated against observations? Why the significant complexity? If used operationally, will your changes (multi-phase, Wang) be incorporated operationally? This should probably be added to the results section.

[L184] "GEM-MACH simulations were carried out on a limited area model (LAM) domain with 2.5 km × 2.5 km (red) resolution, nested from a 10 km × 10 km (blue) horizontal resolution, for the months of April and July, 2018" – refer to figure 1 here

[L200-L265] – I like the fact that you name the simulation here: "base-case", "multi-phase", "Wang2014". Please use these names consistently throughout the manuscript to identify which simulations you refer to. For example, in the supplement in the caption for Fig. S2 you say "partitioned and base experiments (e.g. rain/snow – base)" which is confusing and unnecessary when you have well defined names for these simulations

[L212-L238] Sentences beginning "In both the base-case…" and ending "solid phase precipitation at high altitudes" – This should not be in the description of the simulation but in a separate paragraph after to enhance readability

[L241] "similar to case 2 above" – this is an example of where the simulation name should be used. "case 2" is ambiguous, please instead say "multi-phase". Capitalization of the simulation names ("BASE", "MULTI", "WANG2014") may also increase readability. The actual BCS model could then be referred to as wang2014 to distinguish from the simulation which is a separate entity

[L242-L267] – Sentences beginning "Figure 3 compares the Slinn1984 and Wang2014 …" and ending "where the differences are up to 30% (Wang et al. 2014)". Similar to above, this should not be in the description of the simulation but in a separate paragraph, and probably in the BCS model description (section 2.2) rather than the simulation description (section 2.3)

[L254] – "Jones et al (2022) showed that the thermophoresis mostly enhances the collection of accumulation mode particles (0.1 – 1 µm)" – Jones et al (2022) primarily showed that rear-capture was the dominant BCS process for much of the raindrop size distribution. Similar to the results of this study, Jones et al found little difference in aerosol burdens when using Wang and Slinn, but a large difference between explicit Slinn + phoresis + rear capture and Wang or Slinn. Would your results have been different if you had used the Slinn+ph+rc scheme rather than Wang as the improved model?

[L264] – "The assumptions result in a smaller range of changes for both rain and snow scavenging values as a function of size, generally within 10% for all particle sizes except for particle within 0.1 µm - 2.0 µm for rain scavenging, where the differences are up to 30% (Wang et al. 2014)" – I'm not sure what you mean - the scavenging coefficients seem to differ by orders of magnitude with size not 10-30%?

[L313] – "HSO3- deposition mostly occurs close to the emission sources, while the wet deposition of the oxidized form, SO4= is the dominant and more efficient in downwind regions" – You have not presented maps of emissions and so it is unclear to the uninformed where in the domain is a source and where is a downwind region. I would suggest as a matter of course that in the supplement you plot the accumulated emissions for each of the species for April / July and then refer to these figures when you mention sources/downwind regions etc.

[L328] "Figure 8 indicates enhancement of the scavenged sulphate particles …" – this is certainly true in most but not all regions. Consider saying "overall enhancement"

[L337] "The 90% confidence intervals show …" Please be more specific about the exact statistical test used. I assume a t-test but this is not mentioned

[L349] "Overall, the Wang2014 scheme has slightly lower HSO3 - caused by the feedback in the model, and mixed changes of SO4=, NO3 - and NH4 +. July" – are these overall changes significant? Please call quantify the domain mean change in these components between the different simulations

[L352] Sentence beginning "Here, the regions where the differences are significant…" -  is there a correlation between the deposition anomalies and the precipitation rate? surely this would be easy to determine? Perhaps calculating an $R^2$ between the spatial maps (see here for a method https://agupubs.onlinelibrary.wiley.com/doi/abs/10.1029/1998WR900018)

[L358] "The lower scavenging of the Slinn's scheme can be explained by its lack of processes such as thermophoresis, which may increase the collection efficiency for particles in the size range of 0.01–1 μm (Jones et al., 2022). This may also explain the underestimation of scavenging coefficient from the Slinn (1984) scheme, and the differences between two schemes for particles below 1 μm (refer to section 2.3)." – there is a lot of repetition here about Slinn missing important processes in the Greenfield gap. Please consider condensing this

[L362] – "Figure S4b indicates the difference between two scavenging schemes." Replace 'indicates' with 'shows' or 'highlights' and note this is for snow rather than rain. Please be clearer and more specific with your descriptions of e.g., figures and results as currently it is rather ambiguous throughout the manuscript

[L364] – "Given the fact that the solid precipitation is dominating in the April precipitation" – specifically refer to Figure 2 here

[L376] – "Comparison of the observed SO4= data with the simulation results (Fig. 12d-f), suggests a better agreement with observations by including the Multiphase rain-snow partitioning, and further improvement in agreement associated with the use of the Wang et al. (2014) scavenging scheme" – to the blind eye, the changes are absolutely minimal. Please include goodness of fit metrics in Fig. 12 (NMB, R2, RMSE, etc) to quantify if there is any improvement as without including these metrics, it is difficult to validate this assertion. If the goodness of fit metrics are in the tables, then refer to them in this sentence as well as Fig 12. Fig 12 alone does not show better agreement

[L380] "(from 0.46 to -0.05)" - for SO2! Please be more careful with what you refer to. The lack of description is jarring

[L381] "Wang2014 experiment has a better correlation (R = 0.86) and better factor 2 (Fac2 = 0.64) values compared to the other two GEM-MACH experiments (Table 1)" – without the baseline correlation and factor 2 scores, it is difficult to gauge whether Wang2014 is an improvement. Include the scores for the other 2 experiments here. Are the differences significant or is it in the noise? If you had run a different case study, would you expect to see the same results? See also [L402] for a similar lack of values in the base case which would aid comparability. I would argue that your concluding remark "Overall, the Wang2014 simulation has superior performance to the base case and multiphase Slinn1984 simulations" is only valid if you directly compare the goodness of fit metrics between the simulations. Additionally, I would argue against your assertion given that the NMB is much worse for Wang2014 for SO4 than for the base case for CAPMON!

[L407] – "The impacts of partitioning and Wang2014 scavenging on modelled ambient concentration of speciated PM2.5" – you now move from ions in rain water (deposition) to near-surface concentrations. I assume they are near-surface, please can you clarify this. Please be more careful when describing the metrics, especially in the figures and their captions. It is difficult to determine at present whether the figures show deposition rates or near surface concentrations as this is lacking from the captions

[L412] – "Corresponding 90% confidence interval scores for the difference plots are shown in the lower panels." – I really like the fact that you include the goodness of fit metrics in Figure 16 – I would prefer that a similar thing was done for Fig 12 to aid visual presentation rather than having the values separately in a table but this is just a suggestion. I also like that in Figs 13-15 that the 90 % CI spatial maps are included alongside the anomalies and wonder why you did not do this for Fig 10 and 11 to aid visual presentation

[L438] "For example, the multi-phase approach resulted in the most significant improvement in modelled SO4= wet deposition flux over Alberta (at APQMP sites, and in comparison to previously published work which had wet sulphate positive biases of +200% across combined CAPMoN and APQMP sites, Makar et al., 2018), as well as improvement in modelled ambient particulate sulfate concentration at NAPS sites" – this is my biggest contention with this paper, I don't think that the conclusions are well supported by the results. For example, from table 1, the SO4 NMB is better compared to APQMP but significantly worse compared to CAPMoN. The changes in R and FAC2 are marginal at best. The conclusion should be more conservative I feel – are any of the results actually significant?

[L447] "The Wang et al., (2014) scheme is based on a semi-empirical approach, providing an overall best fit to an ensemble of existing parameterization and observations." – the phraseology is wrong here. Wang certainly performed a best fit optimisation to some existing models but not to observations. They fit their model to the 90% of the parameterizations – an arbitrary choice meant to emphasize that the upper end of the models best fit with observations. However, when you actually compare the Wang-derived scattering coefficients against Laakso "observations" (see Fig. 3 in Jones et al 2022) there remains a significant disparity between Wang and observations. I think this should be highlighted.

[Figure 1] Longitude coordinates are included for the grid cells but latitude coordinates are not – please include these

[Figure 2] why do you say "at the model hybrid level of 0.98"? This is ambiguous – is it near the surface or high in the atmosphere? You also refer to (2c) when I think you mean (2d)

[Figure 5 and 6] – The captions are not informative enough. Do you mean concentration in rainwater? Are these accumulated totals over the month (mass per area) or fluxes (mass per area per time)? Is it normalised to nitrogen and sulfur totals (e.g., mass[S] per area) or in units of substance mass? In the text ([L310]) it is implied that these are contributions to mean wet deposition which is rather ambiguous. Additionally, why do you use µmoles for deposition and then µg for concentration – I would stick to one or the other?

[Figure 7] I am confused as to why you use the units of m rather than mm for the precipitation accumulations? The values range from ~5e-5 to ~7e-3 so mm would be more appropriate. I assume this is an accumulation rather than a rate (which would have units of per time)

[Figure 8] as with figures 5 and 6, the metric being plotted is not described (i.e., difference in accumulated aerosol deposition). Also, please label the subplots (a,b,c, etc) in all of the figures

[Figure 12] Please include goodness of fit metrics in Fig. 12 (NMB, R2, RMSE, etc) to quantify if there is any improvement as without including these metrics, it is difficult to validate this assertion

[Figure 13-15] – I like these figures. The caption is clear and goodness of fit is included in the lower panels

[Tables 1 and 2] Please say what metric is being evaluated in these tables (accumulated deposition for 1, near-surface concentration for 2)

---

## Author Comment (AC1)

*We thank the reviewers for their comments, which have led to improvements of our manuscript. We believe that we have addressed all the comments/concerns. Our point-by-point responses are in blue and Italic font below. Revised texts are highlighted in yellow in the updated manuscript.*

**General Comments**

1. The abstract is too wordy and does not follow the convention of describing the problem, then the method of solving the problem, and then the key results.

*Revised Abstract. Below-cloud scavenging (BCS) is the process of aerosol removal from the atmosphere between cloud-base and the ground by precipitation (e.g. rain or snow), and affects aerosol number/mass concentrations, size distribution, and lifetime. An accurate representation of precipitation phases is important in treating BCS as the efficiency of aerosol scavenging differs significantly between liquid and solid precipitation. The impact of different representations of BCS on existing model biases was examined through implementing a new aerosol BCS scheme in the Environment and Climate Change Canada (ECCC) air quality prediction model GEM-MACH and comparing with the existing scavenging scheme in the model. Further, the current GEM-MACH employs a single-phase precipitation for BCS: total precipitation is treated as either liquid or solid depending on a fixed environment temperature threshold. Here, we consider co-existing liquid and solid precipitation phases as they are predicted by the GEM microphysics. GEM-MACH simulations, in a local-area domain over the Athabasca oil sands areas, Canada, are compared with observed precipitation samples, with a focus on the particulate base cation $NH_4^+$, acidic anions $NO_3^-$, $SO_4^=$, $HSO_3^-$ in precipitation, and observed ambient particulate sulphate, ammonium and nitrate concentrations.*

*Overall, the introduction of the multi-phase approach and the new scavenging scheme enhances GEM-MACH performance compared to previous methods. Including multi-phase approach leads to altered $SO_4^{2-}$ scavenging and impacts the BCS of $SO_2$ into the aqueous phase over the domain. Sulphate biases improved from +46% to -5% relative to Alberta Precipitation Quality Monitoring Program wet sulphate observations. At Canadian Air and Precipitation Monitoring Network stations the biases became more negative, from -10% to -30% for the tests carried out here. These improvements contrast with prior annual average biases of +200% for $SO_4^=$, indicating enhanced model performance. Improvements in model performance (via scores for correlation coefficient, normalized mean bias, and/or fractional number of model values within a factor of two of observations) could also be seen between the base-case and the two simulations based on multi-phase partitioning for $NO_3^-$, $NH_4^+$, and $SO_4^=$. Whether or not these improvements corresponded to increases or decreases of $NO_3^-$ and $NH_4^+$ wet deposition varied over*

*the simulation region.  The changes were episodic in nature – the most significant changes in wet deposition were likely at specific geographic locations and represent specific cloud precipitation events. The changes in wet scavenging resulted in a higher formation rate and larger concentrations of atmospheric particle sulphate.*

2.  Why was the Wang et al (2014) model chosen to replace Slinn (1984) rather than explicitly adding the missing processes of phoreses, electric charge and rear capture which has been shown to better represent observations and empirical models than both Slinn and Wang (see e.g., Fig 3 in Jones et al., 2022)?

    *The decision to incorporate the Wang et al. (2014) model was made considering the specific objectives and constraints of our work, which is part of the ongoing development of the GEM-MACH model at Environment and Climate Change Canada (ECCC). While we acknowledge the potential benefits of including additional processes such as phoresis, electric charge, and rear capture, our study focused on evaluating the impact of the proposed modifications within the scope of our current model framework. The choice of the Wang et al. (2014) model was motivated by its compatibility with the existing version of GEM-MACH. It is important to note that our study represents an incremental step in the model's development, and we recognize the potential for further improvements by incorporating advanced scavenging processes in future studies.*

    *L99 - "Therefore, the Slinn parameterization misses important processes in the Greenfield gap, such as thermophoresis and electrostatic forces, which are included implicitly in the Wang et al. (2014) model. The semi-empirical Wang 2014 scheme was developed to provide an optimization of all available theoretical formulations of scavenging coefficients in comparison with available observations at the time."*

3.  Figures presenting the results in LAM domain (Figs 5-11 and S1-S3) would significantly benefit from having domain-average values also presented. At present, the colour scale and the significant inner-domain variability means it is very difficult to identify the overall impact of changes – e.g., whether the net difference between Wang2014 and multi-phase in Fig. 10 b-d is negative or positive.

    *Thank you for your suggestion. We believe our approach offers a comprehensive view of the impacts in different regions within the domain. Including domain-average values could mask significant local changes and their net effects. However, to address the reviewer comment, we have included the average values for Figs 5-11 and S2-S3.*

4. The figure and table captions are in general not descriptive enough (in some places, outright confusing) and should at the least include the specific metric plotted (e.g., accumulated aerosol deposition in Figs 5 and 6 which are currently erroneously labelled as concentrations). This should also be done for figures in the supplement.

- [Figure 1] Longitude coordinates are included for the grid cells, but latitude coordinates are not. Please include these.
  *Latitude coordinates are now added.*

- [Figure 2] why do you say "at the model hybrid level of 0.98"? This is ambiguous – is it near the surface or high in the atmosphere? You also refer to (2c) when I think you mean (2d).
  *The caption is now corrected, and the hybrid level 0.98 is defined in **L225**: "the model hybrid level of 0.98 (e.g. the level near the surface and corresponding to 98% of total atmospheric pressure)."*

- [Figure 5 and 6] – The captions are not informative enough. Do you mean concentration in rainwater? Are these accumulated totals over the month (mass per area) or fluxes (mass per area per time)? Is it normalised to nitrogen and sulfur totals (e.g., mass[S] per area) or in units of substance mass? In the text ([L310]) it is implied that these are contributions to mean wet deposition which is rather ambiguous. Additionally, why do you use μmoles for deposition and then μg for concentration – I would stick to one or the other?
  *The caption is now corrected. For the average values, we included both units.*

- [Figure 7] I am confused as to why you use the units of m rather than mm for the precipitation accumulations? The values range from ~5e-5 to ~7e-3 so mm would be more appropriate. I assume this is an accumulation rather than a rate (which would have units of per time).
  *The unit is now changed to mm. It is daily accumulated precipitation (PR), averaged over April and July 2018.*

- [Figure 8] as with figures 5 and 6, the metric being plotted is not described (i.e., difference in accumulated aerosol deposition). Also, please label the subplots (a,b,c, etc) in all of the figures.
  *The caption is now corrected.*

- [Figure 12] Please include goodness of fit metrics in Fig. 12 (NMB, R2, RMSE, etc) to quantify if there is any improvement as without including these metrics, it is difficult to validate this assertion.

  *Done.*

5. Please choose informative names for the simulations and then use these names consistently throughout the manuscript.

   *We have addressed the concern regarding the naming of simulations and ensured consistency throughout the manuscript. 1. base-case, 2. multi-phase, and 3. Wang2014.*

6. My biggest issue with the paper is that it does not convincingly show that replacing the single-phase Slinn BCS scheme with the multi-phase Wang BCS scheme significantly improves aerosol concentrations relative to observations, which is explicitly stated as the main result of the paper in both the conclusions and the abstract. Case in point, the abstract includes the line: "Improvements in model performance (via scores for correlation coefficient, normalized mean bias, and/or fractional number of model values within a factor of two of observations) could also be seen, between the base case and the two simulations based on multiphase partitioning for NO3-, NH4+, and SO42." This is patently not true – the paper shows no improvements for NO3 or NH4 (e.g., no areas with significant changes in Fig 9c,d and insignificant improvements to only some skill scores in Tables 1 and 2). I think it is fair to say there is a significant difference in SO4 concentrations between the simulations, but in terms of improvement, SO4 is improved against APQMP observations and made worse against CAPMoN observations in terms of bias, while changes to R2 and FAC2 are marginal. I recommend that the authors be clearer about the significance of their results and include less generalised comments about how the model is improved.

   *Abstract and conclusions are revised now.*

**Specific Comments**

[L1] The title is different to the title in the Supplement. I think the more succinct title here is the more appropriate.

*We changed the title for the supplement.*

[L9] It is unclear what you mean by distributions – aerosol size distributions? Please be more descriptive.

*L8 - We added "size distribution".*

[L11] References to Makar et al (2018) are unnecessary in the abstract. Consider also removing references to Slinn (1984) and Wang et al (2014).

*We removed the references in the abstract.*

[L13] Please give the long-name for GEM-MACH here.

*Done.*

[L16] "GEM-MACH simulations…" – please be a little bit more specific about the experiment design (e.g., Regional GEM-MACH simulations in a local-area domain over Canada).

*L16 - "in a local-area domain over the Athabasca oil sands, Canada".*

[L17] It would help to define the chemical formulae at the outset, e.g., sulfate (SO42=) and then use the chemical formulae or the chemical name consistently throughout the paper. Bouncing between formulae and names is confusing to the reader who might think that one refers to particulates and the other is in reference to gaseous component.

*We have used the chemical formulas and names for the ions (wet deposition) and particles, respectively.*

[L31] Sentences beginning "The aerosol scavenging rates…" and ending "…bigger differences for aerosols larger that 1um" – these are very technical statements for an abstract and should probably be moved to the model description or results. They do not add anything to the abstract. Instead, earlier in the abstract you could describe the main difference between Wang and Slinn in one sentence or at least the motivation for moving from Slinn to Wang.

*We have removed the statements from the abstract. The motivation is discussed later in the model description.*

[L47] – "though the relative importance of aerosol dry deposition may have increased as a result of new observational studies" – the statement is rather awkward and could be interpreted that aerosol dry deposition was directly influenced by the observational studies. Please reword.

*L46 - "however, it is worth noting that recent observational studies, such as Emerson et al., 2020, highlighted the significance of aerosol dry deposition."*

[L49] – by 'precipitation chemistry' do you mean the chemical reactions within the hydrometeor. This phrase is rather ambiguous.

*We clarified the sentence. L48 - "In general, the study of the wet deposition process requires an understanding of cloud processes, including the chemical reactions occurring within hydrometeor."*

[L50] – when you list the microphysical processes involved in BCS you could include the rear-capture effect. I think it is well understood that rear-capture in the wake of an oblate droplet is an important BCS process (e.g., https://acp.copernicus.org/articles/17/4159/2017/) and the reason it is not often considered is that the original BCS studies considered droplets to be spherical.

*Thanks – we included rear capture effect in the list and referred to the paper.*

[L60] You say "below-cloud wet scavenging" or "below-cloud scavenging" a lot, consider acronymising this to make the manuscript easier to read.

*We replaced "below-cloud scavenging" with BCS.*

[L67] "while rain droplets are usually assumed to be spherical in shape" – it is true that most of the early BCS studies assumed spherical raindrops. It would be worth noting that this assumption is erroneous, particularly for larger raindrops, which are much more oblate.

*We added more information. L68 - "However, it is important to note that this assumption can introduce inaccuracies. This is particularly evident for larger raindrops, which often deviate from perfect spherical shapes and exhibit more oblate forms."*

[L79 – L87] – this really doesn't belong in the abstract, please move it to the methodology section (2.2). This describes the underlying BCS scheme and does not fit well in the abstract.

*Thank you for your feedback. While I understand your perspective, the intention behind including the information in the introduction was to establish a foundational understanding of the key concepts related to below-cloud scavenging and the associated equations to explain the study.*

[L87] I find it curious that the extinction efficiency $E(dp, Dd)$ which dominates the variation in the scavenging coefficient is not described in more detail here. Potentially you could say "the Slinn (1984) extinction efficiency is prescribed as a linear combination of the extinction efficiencies due to Brownian motion, interception, and impaction. Therefore, the Slinn model misses important processes in the Greenfield gap such as thermophoresis which are included implicitly in the Wang et al (2014) model. This omission forms the motivation for testing the Wang scheme in this study".

*Thanks for the note. it is added now: **L97** - "The default GEM-MACH scavenging scheme is based on Slinn 1984, and its collection efficiency is prescribed as a linear combination of the collection efficiencies due to Brownian motion, interception, and impaction. Therefore, the Slinn parameterization misses important processes in the Greenfield gap, such as thermophoresis and electrostatic forces, which are included implicitly in the Wang et al. (2014) model. This omission forms the motivation for testing the Wang et al. (2014) scheme in this study."*

[L87] – Are the extinction efficiencies calculated offline or online? If offline, what parameters and assumptions were made to generate the extinction efficiency (temperature, pressure, etc)? What model was used for the raindrop terminal velocity? These are important assumptions which will affect the results.

*The collection efficiencies are calculated online. For rain scavenging, the mean droplet radius is parameterized based on precipitation rate; for snow scavenging, the characteristic length of snow particles and the characteristic capture length are prescribed based on temperature range from Gong et al. (1997). Hydrometeor terminal velocity is parameterized based on Beard (1976). Beard, K.V., 1976. Terminal velocity and shape of cloud and precipitation drops aloft. J. Atmos. Sci. 33 (5), 851–864. Gong, S. L., L. A. Barrie, and J.-P. Blanchet, Modeling sea-salt aerosols in the atmosphere, 1, Model development, J. Geophys. Res., 102, 3805 – 3818, 1997.*

[L94] – What is the motivation for testing Wang et al (2014) over Slinn? At present, it seems arbitrary.

*L97* - "The default GEM-MACH scavenging scheme is based on Slinn 1984, and its collection efficiency is formulated as a linear combination of the collection efficiencies due to Brownian motion, interception, and impaction. Therefore, the Slinn parameterization misses important processes in the Greenfield gap, such as thermophoresis and electrostatic forces, which are included implicitly in the Wang et al. (2014) model. The semi-empirical Wang 2014 scheme was developed to provide an optimization of all available theoretical formulations of scavenging coefficients in comparison with available observations at the time."

[L95] – referring to GEM-MACH, typically the acronym rather than the long name is in parentheses. Also, this should be defined at the first use of the model's acronym.

*Done.*

[L112] "fully-coupled" with reference to GLMs or LAMs typically refers to atmosphere-ocean coupling unless otherwise specified, please be more specific about what is fully coupled to what

*L116 - fully-coupled here refers to the aerosol chemistry and meteorology coupling.*

[L130] Sentence beginning "The default GEM-MACH model includes…" - This seems like a very comprehensive aerosol and chemistry model. Has it been used for other simulations - is it validated against observations? Why the significant complexity? If used operationally, will your changes (multi-phase, Wang) be incorporated operationally? This should probably be added to the results section.

*GEM-MACH is a comprehensive aerosol and chemistry model, and it is used operationally. We are currently testing the inclusion of multi-phase and Wang scheme for operational GEM-MACH.*

[L184] "GEM-MACH simulations were carried out on a limited area model (LAM) domain with 2.5 km × 2.5 km (red) resolution, nested from a 10 km × 10 km (blue) horizontal resolution, for the months of April and July, 2018" – refer to figure 1 here.

*Done.*

[L200-L265] – I like the fact that you name the simulation here: "base-case", "multi-phase", "Wang2014". Please use these names consistently throughout the manuscript to identify which simulations you refer to. For example, in the supplement in the caption for Fig. S2 you say "partitioned and base experiments (e.g. rain/snow – base)" which is confusing and unnecessary when you have well defined names for these simulations.

*Thanks for the note - We have addressed the concern regarding the naming of simulations and ensured consistency throughout the manuscript. 1. base-case, 2. multi-phase and 3. Wang2014.*

[L212-L238] Sentences beginning "In both the base-case…" and ending "solid phase precipitation at high altitudes" – This should not be in the description of the simulation but in a separate paragraph after to enhance readability.

*We made it a separate paragraph starting **L216**.*

[L241] "similar to case 2 above" – this is an example of where the simulation name should be used. "case 2" is ambiguous, please instead say "multi-phase". Capitalization of the simulation names ("BASE", "MULTI", "WANG2014") may also increase readability. The actual BCS model could then be referred to as wang2014 to distinguish from the simulation which is a separate entity.

*We have addressed the concern regarding the naming of simulations and ensured consistency throughout the manuscript. 1. base-case, 2. multi-phase and 3. Wang2014.*

[L242-L267] – Sentences beginning "Figure 3 compares the Slinn1984 and Wang2014 …" and ending "where the differences are up to 30% (Wang et al. 2014)". Similar to above, this should not be in the description of the simulation but in a separate paragraph, and probably in the BCS model description (section 2.2) rather than the simulation description (section 2.3).

*We made it a separate paragraph. Section 2.2 is model description in general, and this information is moved to separate paragraphs after describing the experiments in section 2.3.*

[L254] – "Jones et al (2022) showed that the thermophoresis mostly enhances the collection of accumulation mode particles (0.1 – 1 µm)" – Jones et al (2022) primarily showed that rear-capture was the dominant BCS process for much of the raindrop size distribution. Similar to the results of this study, Jones et al found little difference in aerosol burdens when using Wang and Slinn, but a large difference between explicit Slinn + phoresis + rear capture and Wang or Slinn. Would your results have been different if you had used the Slinn+ph+rc scheme rather than Wang as the improved model?

*Perhaps we will see different results by using the Slinn+ph+rc scheme rather than Wang.*

[L264] – "The assumptions result in a smaller range of changes for both rain and snow scavenging values as a function of size, generally within 10% for all particle sizes except for particle within 0.1 µm - 2.0 µm

for rain scavenging, where the differences are up to 30% (Wang et al. 2014)" – I'm not sure what you mean - the scavenging coefficients seem to differ by orders of magnitude with size not 10-30%?

*We edited the sentence.*

[L313] – "HSO3- deposition mostly occurs close to the emission sources, while the wet deposition of the oxidized form, SO4= is the dominant and more efficient in downwind regions" – You have not presented maps of emissions and so it is unclear to the uninformed where in the domain is a source and where is a downwind region. I would suggest as a matter of course that in the supplement you plot the accumulated emissions for each of the species for April / July and then refer to these figures when you mention sources/downwind regions etc.

*We agree with the reviewer that the "source" versus "downwind" region needs to be better defined. Most of the $SO_4^=$ and $HSO_3^-$ in the region originates in emissions of $SO_2$ from the large stacks in the Oil Sands area. However, their influence when plotted as emissions is not easy to discern, since the relatively high emissions levels occur only in a few model grid cells (those in which the stack sources are located). **L307** - $HSO_3^-$ deposition mostly occurs close to the $SO_2$ emission sources as it is associated with wet scavenging of gas phase $SO_2$, while the wet deposition of the oxidized form, $SO_4^=$, extends to a broader area downwind from the emission sources. Shown in Fig. S2 are maps of modelled average $SO_2$ concentration at the model hybrid level of 0.98 over the region for the periods of our simulations. The "hotspots" of $SO_2$ indicates the locations of major $SO_2$ emission sources in the oil sands area.*

[L328] "Figure 8 indicates enhancement of the scavenged sulphate particles …" – this is certainly true in most but not all regions. Consider saying "overall enhancement".

*We added the word "overall".*

[L337] "The 90% confidence intervals show …" Please be more specific about the exact statistical test used. I assume a t-test but this is not mentioned.

*We added more information. **L323** – "We computed the 90% confidence interval scores for each of the fields examined. The approach follows Makar et al. (2021) and Geer (2016), using a 90 % confidence level in model predictions, with the statistical measures considered different at the 90 % confidence level when the 90 % confidence ranges do not overlap".*

[L349] "Overall, the Wang2014 scheme has slightly lower HSO3 - caused by the feedback in the model, and mixed changes of SO4=, NO3 - and NH4 +. July" – are these overall changes significant? Please call quantify the domain mean change in these components between the different simulations.

*The changes due to the feedback are not significant. We added **L354** – "These changes are not significant (refer to the mean domain values in the figures captions)."*

[L352] Sentence beginning "Here, the regions where the differences are significant…" - is there a correlation between the deposition anomalies and the precipitation rate? surely this would be easy to determine? Perhaps calculating an R2 between the spatial maps (see here for a method https://agupubs.onlinelibrary.wiley.com/doi/abs/10.1029/1998WR900018).

*The discussion here is revised now. **L349** - These changes are not significant (refer to the mean domain values in the figures captions).*

[L358] "The lower scavenging of the Slinn's scheme can be explained by its lack of processes such as thermophoresis, which may increase the collection efficiency for particles in the size range of 0.01–1 μm (Jones et al., 2022). This may also explain the underestimation of scavenging coefficient from the Slinn (1984) scheme, and the differences between two schemes for particles below 1 μm (refer to section 2.3)." – there is a lot of repetition here about Slinn missing important processes in the Greenfield gap. Please consider condensing this.

*Edited: **L357** – "The lower scavenging of the Slinn's scheme can be explained by its lack of processes such as thermophoresis as discussed in section 2.3".*

[L362] – "Figure S4b indicates the difference between two scavenging schemes." Replace 'indicates' with 'shows' or 'highlights' and note this is for snow rather than rain. Please be clearer and more specific with your descriptions of e.g., figures and results as currently it is rather ambiguous throughout the manuscript.

*Edited.*

[L364] – "Given the fact that the solid precipitation is dominating in the April precipitation" – specifically refer to Figure 2 here.

*We added "refer to figure 2 and Fig. S1".*

[L376] – "Comparison of the observed SO4= data with the simulation results (Fig. 12d-f), suggests a better agreement with observations by including the Multiphase rain-snow partitioning, and further improvement in agreement associated with the use of the Wang et al. (2014) scavenging scheme" – to the blind eye, the changes are absolutely minimal. Please include goodness of fit metrics in Fig. 12 (NMB, R2, RMSE, etc) to quantify if there is any improvement as without including these metrics, it is difficult to validate this assertion. If the goodness of fit metrics are in the tables, then refer to them in this sentence as well as Fig 12. Fig 12 alone does not show better agreement.

*We included the goodness of fit metrics in Fig. 12.*

[L380] "(from 0.46 to -0.05)" - for SO2! Please be more careful with what you refer to. The lack of description is jarring.

*Please refer to table 1. Also, the revised text has clearer description.* **L374** *- Comparison of the observed SO$_4^=$ data with the simulation results (Fig. 12d-f), suggests an overall better agreement with observations by including the multi-phase partitioning, and further improvement in agreement associated with the use of the Wang et al. (2014) scavenging scheme. As shown in table 1, the normalized mean bias values of SO$_4^=$ for the multi-phase and Wang2014 experiments are improved compared to the base-case (from 0.46 to -0.05) due to precipitation partitioning, and Wang2014 experiment has the best correlation (R = 0.86, compared to 0.83 for base run and 0.84 for multi-phase) and the best factor 2 score (0.64, compared to 0.57 for both base run and multi-phase) at APQMP sites (Table 1). For the CAPMoN sites, the correlation values for SO$_4^=$ are slightly better for the multi-phase and Wang2014 experiments (R = 0.92 and 0.93), however, the NMB value is smaller for the base experiment (NMB = 0.10, compared to 0.27 and 0.30 for the other two runs).*

[L381] "Wang2014 experiment has a better correlation (R = 0.86) and better factor 2 (Fac2 = 0.64) values compared to the other two GEM-MACH experiments (Table 1)" – without the baseline correlation and factor 2 scores, it is difficult to gauge whether Wang2014 is an improvement. Include the scores for the other 2 experiments here. Are the differences significant or is it in the noise? If you had run a different case study, would you expect to see the same results? See also [L402] for a similar lack of values in the base case which would aid comparability. I would argue that your concluding remark "Overall, the Wang2014 simulation has superior performance to the base case and multiphase Slinn1984 simulations" is only valid if you directly compare the goodness of fit metrics between the simulations. Additionally, I

would argue against your assertion given that the NMB is much worse for Wang2014 for SO4 than for the base case for CAPMON!

*Please refer to the statistical scores in Table 1. Also, the revised text has clearer description. **L374** - Comparison of the observed $SO_4^=$ data with the simulation results (Fig. 12d-f), suggests an overall better agreement with observations by including the multi-phase partitioning, and further improvement in agreement associated with the use of the Wang et al. (2014) scavenging scheme. As shown in table 1, the normalized mean bias values of $SO_4^=$ for the multi-phase and Wang2014 experiments are improved compared to the base-case (from 0.46 to -0.05) due to precipitation partitioning, and Wang2014 experiment has the best correlation (R = 0.86, compared to 0.83 for base run and 0.84 for multi-phase) and the best factor 2 score (0.64, compared to 0.57 for both base run and multi-phase) at APQMP sites (Table 1). For the CAPMoN sites, the correlation values for $SO_4^=$ are slightly better for the multi-phase and Wang2014 experiments (R = 0.92 and 0.93), however, the NMB value is smaller for the base experiment (NMB = 0.10, compared to 0.27 and 0.30 for the other two runs).*

[L407] – "The impacts of partitioning and Wang2014 scavenging on modelled ambient concentration of speciated PM2.5" – you now move from ions in rain water (deposition) to near-surface concentrations. I assume they are near-surface, please can you clarify this. Please be more careful when describing the metrics, especially in the figures and their captions. It is difficult to determine at present whether the figures show deposition rates or near surface concentrations as this is lacking from the captions.

*We added "near the surface" to the text and Figures captions.*

[L412] – "Corresponding 90% confidence interval scores for the difference plots are shown in the lower panels." – I really like the fact that you include the goodness of fit metrics in Figure 16 – I would prefer that a similar thing was done for Fig 12 to aid visual presentation rather than having the values separately in a table but this is just a suggestion. I also like that in Figs 13-15 that the 90 % CI spatial maps are included alongside the anomalies and wonder why you did not do this for Fig 10 and 11 to aid visual presentation.

*We included the goodness in figure 12 too. For figure 10, we have 4 different fields, and we included the 90% confidence level in a separate figure.*

[L438] "For example, the multi-phase approach resulted in the most significant improvement in modelled SO4= wet deposition flux over Alberta (at APQMP sites, and in comparison to previously

published work which had wet sulphate positive biases of +200% across combined CAPMoN and APQMP sites, Makar et al., 2018), as well as improvement in modelled ambient particulate sulfate concentration at NAPS sites" – this is my biggest contention with this paper, I don't think that the conclusions are well supported by the results. For example, from table 1, the SO4 NMB is better compared to APQMP but significantly worse compared to CAPMoN. The changes in R and FAC2 are marginal at best. The conclusion should be more conservative I feel – are any of the results actually significant?

*L433 - For example, the multi-phase approach resulted in the most significant improvement in modelled SO$_4^=$ wet deposition flux over Alberta (at APQMP sites, reducing NMB from 0.46 to -0.05), as well as improvement in modelled ambient particulate sulfate concentration at NAPS sites.*

*Also, the context (discussion) included in L374-380 have provided in discussing the improvement in model results in this study as compared to the previous evaluation in Makar et al. (2018) in 3.2.*

[L447] "The Wang et al., (2014) scheme is based on a semi-empirical approach, providing an overall best fit to an ensemble of existing parameterization and observations." – the phraseology is wrong here. Wang certainly performed a best fit optimisation to some existing models but not to observations. They fit their model to the 90% of the parameterizations – an arbitrary choice meant to emphasize that the upper end of the models best fit with observations. However, when you actually compare the Wang-derived scattering coefficients against Laakso "observations" (see Fig. 3 in Jones et al 2022) there remains a significant disparity between Wang and observations. I think this should be highlighted.

*L440 – The Wang et al. (2014) scheme is based on a semi-empirical approach, and implicitly accounts for electrostatic forces, which are shown to be more important than diffusiophoresis in Jones et al. (2022). This scheme provided an overall best fit to an ensemble of existing models, although there is still a significant disparity between the scavenging coefficients based on Wang et al. (2014) and some of the observation-based scavenging coefficients (e.g., Jones et al. 2022).*

---

## Author Comment (AC2)

*We thank the reviewer for the comments, which have led to improvements of our manuscript. We believe that we have addressed all the comments/concerns. Our point-by-point responses are in blue and Italic font below. Revised texts are highlighted in yellow in the updated manuscript.*

In this study, the authors considered and implemented different treatments of below-cloud scavenging. Compared to the previous scheme that is based on Slinn (1984), a new scheme that considers multiphase hydrometeors explicitly from the microphysic scheme shows a better agreement with observations. In addition, a semi-empirical model based on Wang et al. (2014) shows a better performance than the previous scheme (Slinn, 1984). Overall, however, the differences are quite small especially for nitrate and ammonium aerosols and their wet depositions. I would recommend adding more supportive analyses or conclusive remarks for NO3 and NH4 species.
Specific comments:

1. Abstract is too long, so please make it more concise. Also, I wonder if the references are necessary in the abstract.
*We have shortened the abstract and removed the references.*

*__Revised Abstract.__ Below-cloud scavenging (BCS) is the process of aerosol removal from the atmosphere between cloud-base and the ground by precipitation (e.g. rain or snow), and affects aerosol number/mass concentrations, size distribution, and lifetime. An accurate representation of precipitation phases is important in treating BCS as the efficiency of aerosol scavenging differs significantly between liquid and solid precipitation.  The impact of different representations of BCS on existing model biases was examined through implementing a new aerosol BCS scheme in the Environment and Climate Change Canada (ECCC) air quality prediction model GEM-MACH and comparing with the existing scavenging scheme in the model. Further, the current GEM-MACH employs a single-phase precipitation for BCS: total precipitation is treated as either liquid or solid depending on a fixed environment temperature threshold. Here, we consider co-existing liquid and solid precipitation phases as they are predicted by the GEM microphysics. GEM-MACH simulations, in a local-area domain over the Athabasca oil sands areas, Canada, are compared with observed precipitation samples, with a focus on the particulate base cation $NH_4^+$, acidic anions $NO_3^-$, $SO_4^=$, $HSO_3^-$ in precipitation, and observed ambient particulate sulphate, ammonium and nitrate concentrations.*

*Overall, the introduction of the multi-phase approach and the new scavenging scheme enhances GEM-MACH performance compared to previous methods. Including multi-phase approach leads to altered $SO_4^{2-}$ scavenging and impacts the BCS of $SO_2$ into the aqueous phase over the domain.  Sulphate biases improved from +46% to -5% relative to Alberta Precipitation Quality Monitoring Program wet sulphate observations.  At Canadian Air and Precipitation Monitoring Network stations the biases became more negative, from -10% to -30% for the tests carried out here. These improvements contrast with prior annual average biases of +200% for $SO_4^=$, indicating enhanced model performance. Improvements in model performance (via scores for correlation coefficient, normalized mean bias, and/or fractional number of model values within a factor of two of observations) could also be seen between the base-case and the two simulations based on multi-phase partitioning for $NO_3^-$, $NH_4^+$, and $SO_4^=$. Whether or not these improvements corresponded to increases or decreases of $NO_3^-$ and $NH_4^+$ wet deposition varied over the simulation region.  The changes were episodic in nature – the most significant changes in wet deposition were likely at specific geographic locations and represent specific cloud precipitation events.*

*The changes in wet scavenging resulted in a higher formation rate and larger concentrations of atmospheric particle sulphate.*

2. I wonder how equation (2) can be connected to the other equations (4–6) that describe the scavenging coefficient.
*Equation (2) represents the scavenging coefficient by precipitation (rain and snow) for various aerosol species. To provide additional clarity, we have changed it to a capital letter, "$\Lambda$".*
*Slinn (1984) separated this equation to two different equations (e.g. 3 and 4) to introduce below-cloud scavenging coefficients for rain and snow, respectively.*

3. Line 203. a "uniform" environmental temperature threshold can be misleading. May be "constant" or "fixed" better?
*Thank you for the input – it has been changed to "constant".*

4. Line 309. "the modeling science" sounds awkward.
*Thank you for the input – The phrase has been replaced with "based on an earlier version of the GEM-MACH mode".*

5. Line 319. I'm not sure if this assertion is right. Particulate nitrate concentrations can be higher in winter than in summer. Considering that HNO3 gas is very soluble, either HNO3 and particulate NO3 would be well scavenged by cloud water. I think the amount of emitted NOx can be higher in winter, so recommend seeing if NOx emissions are higher in winter than in summer.
*Thank you for your comment - While the uptake of higher particulate nitrate concentrations into cloud water is a contributing factor, we acknowledge the influence of elevated NOx emissions in the winter due to increased energy demand. We believe the manuscript reflects this - We also added Line 319 to address this: This is further influenced by the elevated NOx emissions during the winter due to increased energy demand.*

6. Line 335–343. Explanation for sulfate seems straightforward. However, nitrate and ammonium do not. My question is why the model performance for NO3 and NH4 wet deposition is better with the new schemes compared to the old one. As seen in Fig. 8 and Fig. 9, the difference in NO3 and NH4 fluxes vary greatly across regions and their *significant scores are below a 90% confident level over the most of the region.*
*Thank you for your question - For sulfate particles, our model's improvements are consistent because sulfate tends to behave similarly in cloud processes. It dissolves well and gets washed out effectively by rain and snow. However, for nitrate and ammonium, things are a bit more complicated. Their wet deposition and their behavior depends on local conditions like temperature and rainfall rates. This can lead to irregular patterns in their wet deposition. As a result, the differences in wet deposition depend on the nature and size of the particles and how they react to different weather conditions. The confidence intervals in our results reflect these variations.*

7. Line 388–399. I think that a comparison with a previous study, Makar et al. (2018) is not an apple-to-apple comparison. As the authors mentioned, in this study, they updated dry deposition, so the higher dry deposition velocities in this study than in the previous study can be a reason of the better performance of this study. If possible, how about comparing the new below-cloud scavenging schemes to the old one when the same dry deposition scheme is used?

*Thank you for your comment - We can confirm that there were no changes made to the dry deposition scheme in our study compared to Makar et al. (2018). We removed the wrong statement from the manuscript.*

8. I wonder if the better agreements with new schemes and corresponding explanation (reasoning) can be consistently applied to all species (SO4, NO3, and NH4). Also, wonder if the improvements can be due to some wrong reasons.

*Thank you for the comment - Our study shows that the changes we made, e.g. using the multi-phase and Wang et al. (2014) schemes, improves our model's performance. These improvements are especially noticeable when dealing with sulfate wet deposition. The variations we see in nitrate and ammonium wet deposition are not the same everywhere; they come and go with specific weather conditions. When we compare our model's results with real observations, it's evident that the Wang et al. (2014) scheme, along with considering multi-phase approach, provides a better match with actual conditions.*

9. Overall, figures need to be improved to highlight clearly the differences or conclusions that they want to deliver. Please adjust colors or value scales accordingly. Also, I'd recommend combining Fig. 8 and Fig. 9 if possible; for example, you can add hatched lines or stipples over regions only where the score >= 1 on Fig. 8. Same for Fig. 10 and Fig. 11, and Fig. 13–15.

*Thank you for the comment - We've improved our figures based on the first reviewer's suggestions and will upload a new version. We value both reviewers' feedback.*

---

## Author Response (AR2)

Dear Dr. Remy,

We have incorporated the suggested revision to the abstract as per your feedback.

Best regards,
*Roya Ghahreman*, Ph.D.
Scientist,
Air Quality Research Division,
Science & Technology Branch,
Environment and Climate Change Canada (ECCC),
Office: 4S611, 4905 Dufferin Street,
Toronto, ON, M3H 5T4, Canada
roya.ghahreman@ec.gc.ca
roya.ghahreman@utoronto.ca
+1 (416) 739 4690